# Microscale model comparison (benchmark) at the moderate complex forested site Ryningsnäs

Stefan Ivanell[1], Johan Arnqvist[1], Matias Avila[2], Dalibor Cavar[3], Roberto Aurelio Chavez-Arroyo[4], Hugo Olivares-Espinosa[1], Carlos Peralta[5], Jamal Adib[5], and Björn Witha[6]

[1]Uppsala University, Wind Energy Section, Campus Gotland, 621 67 Visby, Sweden
[2]Barcelona Supercomputing Center, BSC, Spain
[3]Wind Energy Department, Technical University of Denmark, Denmark
[4]National Renewable Energy Centre (CENER), Spain
[5]Wobben Research and Development MS GmbH, Germany
[6]ForWind - Carl von Ossietzky Universität Oldenburg, Germany

*Correspondence to:* Stefan Ivanell (stefan.ivanell@geo.uu.se)

This is for general edits This is for answers to referee 1 This is for answers to referee 2

**Abstract.** This article describes a study where modellers were challenged to compute the wind field at a forested site with moderately complex topography. The task was to model the wind field in stationary conditions with neutral stratification by using the wind velocity measured at 100 m at a metmast as only reference. Detailed maps of terrain elevation and forest densities where provided as only inputs, derived from Airborne Laser Scans (ALS) with a resolution of 10 m x 10 m covering an area of 50 km x 50 km, that closely matches the actual forest and elevation of the site. The participants were free to apply their best practices for the simulation, that is, to decide the size of the domain, the value of the geostrophic wind and every other modelling parameter. The comparison of the results with the measurements is shown for the vertical profiles of wind speed, shear, wind direction and turbulent kinetic energy. The ALS based data resulted in reasonable agreement of the wind profile and turbulence magnitude. The best performance was found to be that of Large-Eddy Simulations using a very large domain. For the Reynolds-Averaged Navier-Stokes type of models the constants in the turbulence closure were shown to have a great influence in the yielded turbulence level, but were of much less importance for the wind speed profile. Of the variety of closure constants used by the participating modellers, the closure constants from Sogachev and Panferov (2006) proved to agree best with the measurements. Particularly the use of $C_\mu \approx 0.03$ in the $k$-$\varepsilon$ model obtained better agreement with turbulence level measurements. All exepct two participating models used the full detailed ground and forest information to model the forest which is considered a significant progress compared to previous conventional approaches. Overall, the article gives an overview of how well different types of models are able to capture the flow physics at a moderately complex forested site.

*Copyright statement.* TEXT

# 1 Introduction

To respond to the increasing demand for wind power, new areas for wind turbine siting are being explored. Large off-shore farms further away from the shore are being developed as well as wind farms in more complex on-shore areas, such as terrain with a more varied topography and roughness. This is the case in northern countries, such as the Scandinavian region, where large remote forested areas are being explored for wind development. However, when exploring these complex sites it is evident that new challenges arise due to comparatively higher turbulence levels and wind shear. While the magnitude of wind shear and turbulence increase the fatigue load, uncertainties in the estimation of wind shear and turbulence have shown to be problematic in forested areas Enevoldsen (2016). Hence, it is important to assess the uncertainties in the modelling process of wind conditions over forests.

In addition to the actual difference in wind climate between traditional wind energy sites and complex forested ones, modelling of the wind conditions is challenging. Trees are elevated sources for both momentum absorption and heat transfer and thus they differ from traditional surfaces since the exchange may be distributed at several model levels. The degree of physical description is a choice by the modeller, going from describing Plant Area Densities (PAD) in each grid cell to representing an entire forest by a single roughness length value. The required numerical demand does, however, vary with many orders of magnitude when making that choice.

To the knowledge of the authors, no large-scale studies have been published comparing different micro scale models over forested terrain with high quality meteorological data. However, Ayotte (2008) compared models of varying complexity to wind tunnel measurements and concluded that inaccurate representation of all physical scales may result in significant errors. Earlier model intercomparison studies of micro-scale models in non-forested areas have provided insight to the performance of different model types and highlighted important differences in modelling choices such as closure constants (Bechmann et al., 2011; Bosveld et al., 2014). So far there remains significant uncertainties in how well wind climate models perform in forested areas, and there are also large differences between model descriptions. Hence there is a need for more validation studies and a better understanding of how the different modelling choices affect the final result. This study aims to take the first steps to fill that knowledge gap by presenting model performances at a forested wind turbine site.

The progress of forest flow modelling now enables direct simulation of the tree densities. Such values, PAD, may be derived from Airborne Laser Scans (ALS) that are becoming increasingly available from national mapping services (Boudreault et al., 2015). Using PAD data instead of estimated roughness lengths may be a way to reduce the uncertainties of site assessment. However, the use of drag modelling through PAD is increasingly being adopted in the wind energy community and how to best make use of the data is still an open question, as well as how its use affects the model abilities. In order to the test the performance of the wind simulation determined by models using PAD derived from ALS, data corresponding to this quantity was made available to the modellers taking part in the study.

The study started with a call for a benchmarking model validation study to modellers involved in the European ERANET+ project New European Wind Atlas (NEWA). The aim of the benchmark is to illustrate how well micro-scale models are able to simulate winds above a forest in moderately complex topography. The participating models range from industrial wind models

to frontline research models. The modelled case consists of a typical site located in Ryningnäs in southern Sweden, i.e. a patchy forested site with moderately complex topography Arnqvist et al. (2015). The aim of the benchmark study is in that sense quite broad and aims to answer the question of how successful different modelling programs, as well as modelling practices, are in matching the measured wind profile in specified conditions. The instructions for the benchmark was deliberately kept rather open in order to study how the different groups made use of the PAD data, and how the compromise between resolution and domain size was handled.

The NEWA project includes several large scale field campaigns designed for flow model validation Mann et al. (2017), however, the Ryningnäs measurements campaign was performed prior to the start of NEWA project and was identified as an appropriate dataset for a benchmarking study. As such, it also forms a basis for model validation methodology as preparation for upcoming microscale benchmarks (Mann et al., 2017) using measurement input from the extensive measurements campaigns performed within the NEWA project.

The paper begins by outlining the following; the benchmark, the validation data and general modelling. This is followed by a description of first the RANS models and then the LES models. It then continues with the main results and finally concludes with a discussion and conclusions section.

## 2  Benchmark description

The benchmark task was to model the wind profile at the location $57°16'34.26"$N, $15°59'12.23"$E for the wind directions $100°$, $240°$ and $290°$ (directions at 100 m height). The input provided to the modellers was a target wind speed of 7.4 m/s at 100 m height, neutral atmospheric stratification and a data set of forest density and topography in a 40 km $\times$40 km grid. The modellers were asked to provide the wind profile from the ground up to at least 200 m, geostrophic wind speed $U_g$, wind speed in planes at 40, 100 and 140 m above local ground level (AGL) as well as information about their model.

The choice behind having a target wind speed at 100 m height, rather than having a fixed geostrophic wind speed was connected partly due to the lack of measurements of the geostrophic wind speed and partly due to the desire to have as similar a wind speed as possible in the lower part of the boundary layer. This relates to the question of whether or not the models can accurately predict the flow foot print, given that the ALS data enables the model to have surface conditions very similar to reality. Setting a fixed geostrophic wind speed would risk that the modelled wind speed in the surface layer may end up being lower or higher than the measured, with a subsequent uncertainty of the ability to capture the foot print of the flow. In a strictly neutral boundary layer, the ratio of the turbulence level and the wind speed is expected to be constant, and hence the foot print would be the same for different wind speeds. However, the boundary layer height changes with wind speed as well as the gradients of velocity and turbulence level. Thus, in addition to scaling the wind speed with the friction velocity, one would have to scale the height with the boundary layer height in order to make a fair comparison, which cannot be done since the boundary layer height was not measured.

## 3 Measurements

### 3.1 Forest characterisation by laser scans

In order to characterise the forest ALS data from the Swedish map authority Lantmäteriet has been utilised (Lantmäteriet, 2016). The data was collected at a flight height of 1700 m yielding a foot print area at the ground of 0.5 m$^2$ for the laser beam. The density is around 1 shot/m$^2$. The data was processed according the method described in Boudreault (2015). The method uses the Beer-Lambert law for the attenuation of the laser beam as it travels through the forest canopy and the Plant Area Density (PAD) can be derived if the extinction coefficient is known. The extinction coefficient was here (as in Boudreault, 2015) assumed to be $0.5/cos(\theta_{ALS})$, where $\theta_{ALS}$ is the scanning angle, corresponding to a spherical distribution of canopy elements. The height of the forest was derived by the maximum return height in a grid box as defined by the distance from the median of the returns in that grid box that had been classified as ground (a.k.a. the ground height of the grid box). The PAD was derived in vertical steps of 1 m from the highest return reflection towards the ground. In order to avoid numerical problems when the beam becomes fully attenuated (in very dense forest patches) PAD estimation was set to a constant of 0.1 m$^{-1}$ below the level of 95 % attenuation. Since most of the returns normally comes from the ground, this filter only affected 0.1 % of the PAD estimates. Two data sets of a resolution of 10 m by 10 m and 50 m by 50 m respectively were then prepared to be used as model input. The data sets include horizontal coordinates, ground height, tree height and PAD in a vertical grid from 0 m above ground to the tree height in steps of 1 m.

### 3.2 Site description

The measurement site is located in Ryningsnäs in south-east Sweden (57°16′34.26"N, 15°59′12.23"E). A meteorological measurement tower, equipped with cup and sonic anemometers, with the highest measurement level 138 m above ground has been used for the validation data. The area around the tower site has moderate complexity in terms of topography, but the forest cover is very heterogeneous with many clearings and forest stands of different ages. Figure 1 shows the forest cover on different scales, the largest being 50 km by 50 km and the smallest 1 km by 1 km. The three sectors chosen for the validation study have been coloured.

The tower is situated in the north-west corner of a 400 m ×400 m clearing. The surrounding forest has a peak in the tree height distribution at approximately 20 m and is predominantly consists of Scotts Pine (*Pinus Sylvestris*). The actual tree height distribution can be seen in Fig. 2 $(a)$ and $(b)$ where the distribution is shown for the three sectors within a radius of 10 km $(a)$ and 1 km $(b)$. In the larger scale, the tree height distributions of sectors $100°$ and $240°$ are very similar and have a peak at 21.5 m. The distribution of sector $290°$ is more flat, with two maximums, one at 7.5 m and one at 19.5 m. In the closer region the $290°$ distribution is different from the two other sectors in that it does not contain a clearing, but does contain a large patch of young forest at 7.5 m height. The clearing of the tower itself can be seen in the distribution of the $100°$ sector whereas the clearing in the $240°$ sector is almost one km upwind.

Two wind turbines are situated approximately 200 m to the north east and south of the tower respectively, but the three sectors used in the validation study exclude directions which these turbines would influence.

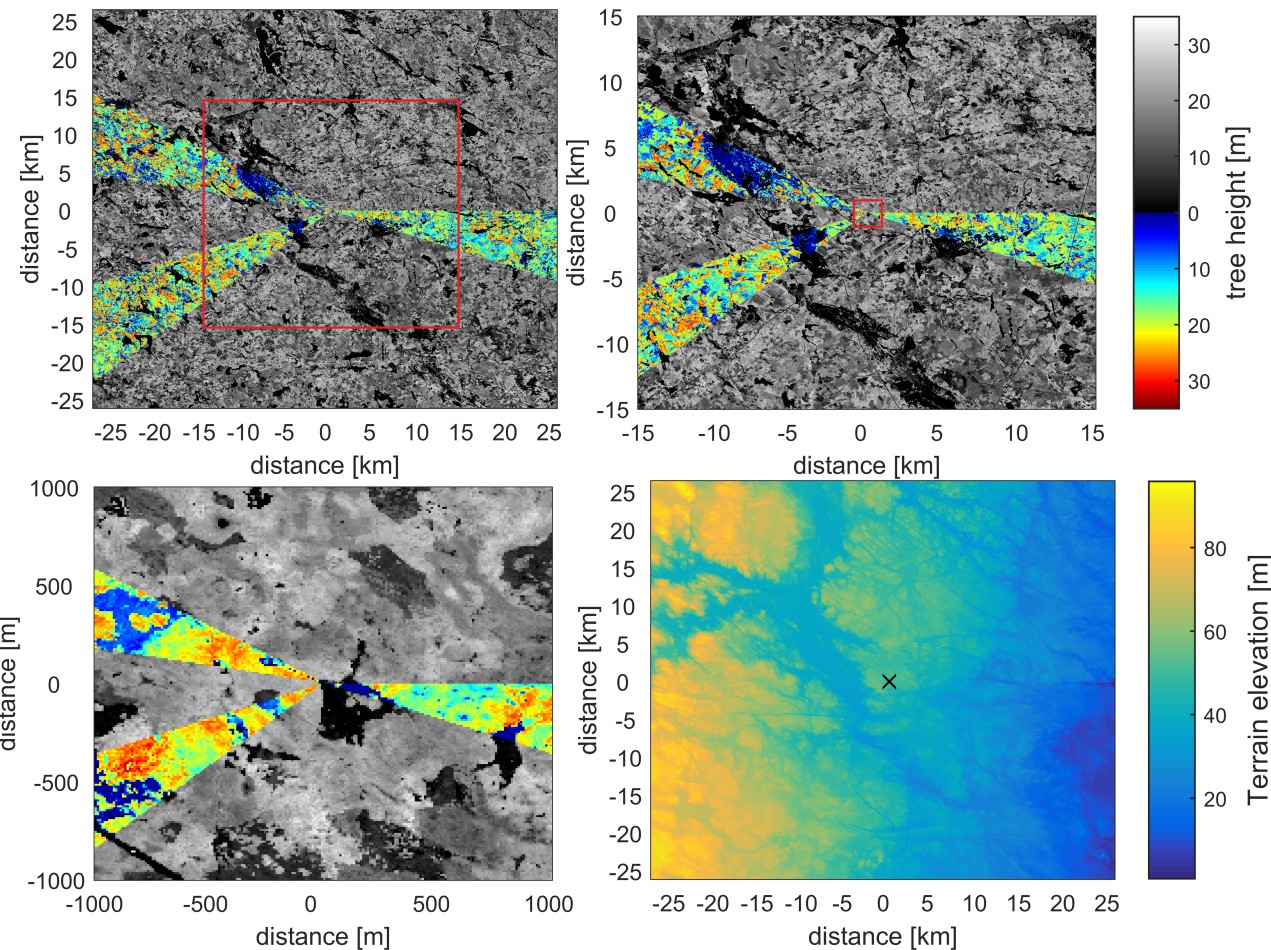

**Figure 1.** $(a) - (c)$ Forest cover of the Ryningsnäs site with increasing magnification factors. The contour maps in grey scale and colour scale illustrate the tree height distribution, both inside and outside the selected area. $(d)$ A topographic map of the Ryningsnäs area.

### 3.3 Wind measurements

The full measurement set up and the wind climate has earlier been reported in Arnqvist et al. (2015). The instruments used in this study includes six sonic anemometers (Metek Gmbh, USA-1) located at the heights of 40, 59, 80, 98, 120 and 137.7 meters as well as 7 Thies first-class cup anemometers at the heights 25.5, 40.1, 60.5, 80.1, 95.85, 120.75 and 137.6 m.

5     The sonic anemometers where sampled at 20 Hz and statistics was evaluated by 30 minute block averaging and 3D rotation of the coordinate system aligning it with the local mean wind direction yielding the wind vector $\bar{u}$, $\bar{v}$, $\bar{w}$ and temperature $\bar{t}$. Flow distortion correction and quality checks were applied as in Arnqvist et al. (2015). The stratification was evaluated by means of $(z - d)/L$, where $z$ is the height, $d$ is the displacement height and $L$, the Obukhov length, was determined as

$$L = -\frac{u_*^3 T}{\kappa g \overline{w't'}}, \tag{1}$$

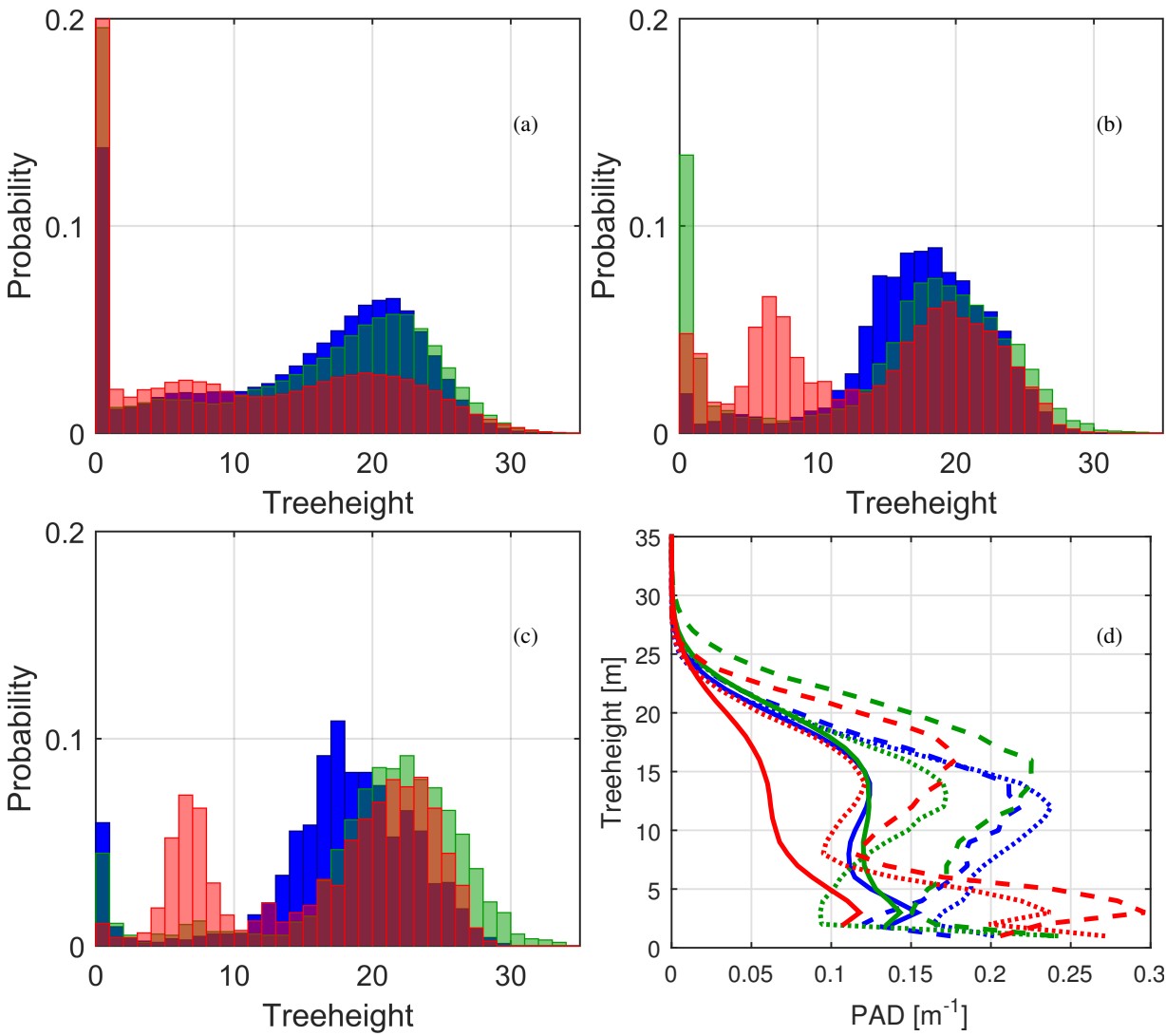

**Figure 2.** (*a*), (*b*) and (*c*) Distribution of tree heights in the three sectors within radius (*a*) 10 km, (*b*) 2 km and (*c*) 1 km. The colours indicate sectors 100° blue, 240° green and 290° red. (*d*), average PAD profiles from within 10 km as full lines, 2 km as dotted lines and within 1 km as dashed lines. Colouring as in the histograms.

where **the velocity fluctuations are defined by the substraction of the mean velocities from the instantaneous values** $u' = u - \bar{u}, v' = v - \bar{v}, w' = w - \bar{w}$; $u_* = (\overline{u'w'}^2 + \overline{v'w'}^2)^{1/4}$, the friction velocity, $\kappa = 0.4$ is the von Kármán constant, $g = 9.81$ m/s, is the gravitational acceleration and $t' = t - \bar{t}$, is the instantaneous temperature fluctuation.

To select only neutral conditions $(z-d)/L$ was required to be between -0.1 and 0.07 at all heights. The limits where selected based on the shape of the $\phi$-function for momentum (Högström, 1996) and allows for roughly a $\pm 35$ % variation in the wind gradient given a certain $u_*$ value. In addition, the 100 m wind speed was required to be between 7 and 8 m/s and the conditions quasi stationary, as defined by that the wind speed was allowed to vary a maximum of 10% between adjacent 10-min segments and the wind direction was allowed to vary a maximum of $10°$ between adjacent 30-min segments. The wind directions were required to be within $\pm 10°$ of the target wind direction. After applying all the conditions and quality controlling the data there remained 9, 13 and 9 separate 30-min segments in the sectors $100°$, $240°$ and $290°$ respectively. To sum up, the data selection consisted of:

- quality check passed

- neutral stratification

- stationary flow

- $(7 < \bar{u}_{98m} < 8)$ m/s

- wind direction within the target sector

The data that fulfilled the neutral and stationary conditions constitute 10 % of the total data set and the ones also fulfilling $(7 < \bar{u}_{98m} < 8)$ m/s make up 1% of the entire data set. Selecting for the three specific directions further reduces the number of data points.

# 4 Modelling

The models that participated in the benchmark where all CFD models using a Reynolds-Averaged Navier-Stokes (RANS) or Large Eddy-Simulation (LES) methodology. Table 1 shows an overview of models used by the respective participant. Some of the models include a full topography and PAD description. All models use a drag formulation to simulate the forest, with a drag coefficient of 0.2 (except Meteodyn which uses another drag formulation, please see Section 4.2.1). In the following section a description of each model setup is given.

## 4.1 General CFD Modelling

Computational modelling of the fluid flow employs a filtered version of the Navier-Stokes equations, due to the impracticality of resolving every temporal and spatial scale. The Reynolds-Averaged Navier-Stokes (RANS) equations make use of the Reynolds decomposition to divide the velocity field into the time averaged velocity and the velocity fluctuation around the

mean, $u_i = \overline{u_i} + u_i'$. This yields a momentum equation for the mean flow where the effect of the turbulent motions is represented via the Reynolds stresses $\overline{u_i' u_j'}$, requiring a model to represent their effect on the average field. Most approaches (known as turbulence viscosity models) employ the Boussinesq approximation where the Reynolds stresses are parameterized as a function of an eddy viscosity $\nu_t$ and a rate of strain tensor, which assumes that the turbulence fluxes are proportional to the mean velocity gradient. This yields,

$$\frac{\partial \overline{u_i}}{\partial t} + \frac{\partial \overline{u_i u_j}}{\partial x_j} = -\frac{1}{\rho}\frac{\partial \overline{p}}{\partial x_i} + \frac{\partial}{\partial x_j}\left[(\nu + \nu_t)\left(\frac{\partial \overline{u_i}}{\partial x_j} + \frac{\partial \overline{u_j}}{\partial x_i}\right)\right] + f_c \varepsilon_{ij3}\overline{u_j} + \overline{f_i}, \tag{2}$$

as the momentum equation, where $f_i$ represents an external body force. The Coriolis force is included using the Coriolis parameter $f_c = 2\Omega \sin\lambda$, where $\Omega$ is the Earth's rotational velocity and $\lambda$ the latitude of the wind farm. The eddy viscosity is modelled through the introduction of transport equations, such as in the frequently employed $k - \varepsilon$ technique.

RANS modelling supposes that the effect of all ranges of fluctuations on the mean flow can be accounted for by the models. Conversely, in the LES approach the energy-containing flow structures are fully resolved, whereas only the effect of the smaller fluctuations is modelled. This is achieved through the decomposition of the velocity field into filtered (or resolved) and residual (or subgrid scale, SGS) components, $u_i = \langle u\rangle_i + u_{\text{SGS}}$. Although various types of filters exist, a very common method in wind research is to associate a filter width to the grid spacing. The application of this decomposition on the Navier-Stokes equations leads to the apparition of the SGS stress tensor $\tau_{ij}^{\text{SGS}} \equiv -\rho(\langle u_i u_j\rangle - \langle u\rangle_i \langle u\rangle_j)$, that needs to be modelled. As in RANS, the prevalent strategy is to apply the Boussinesq approximation to introduce a subgrid viscosity $\nu_{\text{SGS}}$ to derive the LES momentum equation,

$$\frac{\partial \langle u_i\rangle}{\partial t} + \frac{\partial \langle u_i\rangle \langle u_j\rangle}{\partial x_j} = -\frac{1}{\rho}\frac{\partial \langle p_m\rangle}{\partial x_i} + \frac{\partial}{\partial x_j}\left[(\nu + \nu_{\text{SGS}})\left(\frac{\partial \langle u_i\rangle}{\partial x_j} + \frac{\partial \langle u_j\rangle}{\partial x_i}\right)\right] + f_c \varepsilon_{ij3}\langle u_j\rangle + f_i, \tag{3}$$

where $p_m$ denotes the modified pressure, which includes the isotropic part of the SGS stress tensor. The simplest approaches to calculate $\nu_{\text{SGS}}$ (and amongst the most commonly employed in wind research), make use of the resolved scales.

**Table 1.** Overview of used models and model family. [1] PALM used mean values of the ALS derived PAD combined with flat terrain

| | Solver | RANS | LES | ALS input | Closure/SGS | Marker |
|---|---|---|---|---|---|---|
| Meteodyn | Meteodyn WT | x | | | $k - l$ | ∗ |
| EllipSys3D hi res 1 | EllipSys3D | x | | x | $k - \varepsilon$ | ◁ |
| EllipSys3D hi res 2 | EllipSys3D | x | | x | $k - \varepsilon$ | △ |
| EllipSys3D low res 1 | EllipSys3D | x | | x | $k - \varepsilon$ | ▷ |
| EllipSys3D low res 2 | EllipSys3D | x | | x | $k - \varepsilon$ | ▽ |
| CFDWind | OpenFOAM | x | | x | $k - \varepsilon$ | □ |
| UUCGWind | OpenFOAM | | x | x | $k_{SGS}$ | ○ |
| Alya | Alya | x | | x | $k - \varepsilon$ | + |
| PALM spatial average | PALM | | x | x[1] | $k_{SGS}$ | ○ |
| PALM vertical column | PALM | | x | x[1] | $k_{SGS}$ | ◇ |

## 4.2 RANS

The participating RANS models use one or two equations turbulence models, presented in general form in this section. The specific set up of each model is presented in the following sections. The two equations turbulence closure model corresponds to the classical $k - \varepsilon$ model (Launder and Spalding, 1974), where transport equations for the turbulent kinetic energy $k$ and its dissipation rate $\varepsilon$:

$$\frac{\partial k}{\partial t} + \frac{\partial}{\partial x_j}(\overline{u_j}k) - \frac{\partial}{\partial x_j}\left(\left(\nu + \frac{\nu_t}{\sigma_k}\right)\frac{\partial k}{\partial x_j}\right) = \nu_t \frac{\partial \overline{u_i}}{\partial x_j}\left(\frac{\partial \overline{u_i}}{\partial x_j} + \frac{\partial \overline{u_j}}{\partial x_i}\right) - \varepsilon + S_k \tag{4}$$

$$\frac{\partial \varepsilon}{\partial t} + \frac{\partial}{\partial x_j}(\overline{u_j}\varepsilon) - \frac{\partial}{\partial x_j}\left(\left(\nu + \frac{\nu_t}{\sigma_\varepsilon}\right)\frac{\partial \varepsilon}{\partial x_j}\right) = C_{\varepsilon_1}\frac{\varepsilon}{k}\nu_t \frac{\partial \overline{u_i}}{\partial x_j}\left(\frac{\partial \overline{u_i}}{\partial x_j} + \frac{\partial \overline{u_j}}{\partial x_i}\right) - C_{\varepsilon_2}\frac{\varepsilon^2}{k} + S_\varepsilon \tag{5}$$

are solved. $C_{\varepsilon_1}, C_{\varepsilon_2}, \sigma_k, \sigma_\varepsilon$ are modelling constants and $S_k$, $S_\varepsilon$ are the source/sink terms representing the drag based energy loss in the canopy. The eddy viscosity is determined from:

$$\nu_t = C_\mu \frac{k^2}{\varepsilon} \tag{6}$$

where $C_\mu$ is another modelling constant. Following Sogachev and Panferov (2006) the source/sink canopy terms are parametrised as:

$$S_k = 0 \tag{7}$$

$$S_\varepsilon = 12\left(C_{\varepsilon_2} - C_{\varepsilon_1}\right)C_\mu^{\frac{1}{2}}C_d\, a\,|\,\overline{\mathbf{u}}\varepsilon \tag{8}$$

where $C_d$ is a drag coefficient and $a$ is a frontal area density.

The length scale in the standard $k - \varepsilon$ model is not bound and grows indefinitely with height. In order to adjust the model to ABL relevant flow cases, a correction suggested by Apsley and Castro (1997) is applied, where the $C_{\varepsilon_1}^*$ constant is (re)defined in a following manner:

$$C_{\varepsilon_1}^* = C_{\varepsilon_1} + (C_{\varepsilon_2} - C_{\varepsilon_1})\frac{l}{l_{max}}, \tag{9}$$

with the mixing length $l$ defined by:

$$l = C_\mu^{\frac{3}{4}}\frac{k^{\frac{3}{2}}}{\varepsilon} \tag{10}$$

The limiting mixing length scale $l_{max}$ is determined based on the relationship proposed by Blackadar (1962):

$$l_{max} = \frac{0.00027\,G}{f_c} \tag{11}$$

**Table 2.** Summary of the used model constants.

| | Standard $k-\epsilon$ | Modified $k-\epsilon$ | Modified2 $k-\epsilon$ | $k-l$ | LES |
|---|---|---|---|---|---|
| $C_{\epsilon 1}$ | 1.52 | 1.176 | 1.13 | 0 | - |
| $C_{\epsilon 2}$ | 1.833 | 1.920 | 1.9 | 0 | - |
| $C_\mu$ | 0.09 | 0.033 | 0.0256 | 0 | - |
| $C_d$ | 0.2 | 0.2 | 0.2 | - | 0.2 |
| $C_d^* = C_d/A$ | - | - | - | 0.005 | - |
| $\sigma_k 1$ | 1.0 | 1.0 | 0.7407 | 0 | - |
| $\sigma_{\epsilon 1}$ | 1.7039 | 1.238 | 1.2987 | 0 | - |
| Used by | EllipSys3D 1 | Alaya, EllipSys3D 2 | CFDWind | Meteodyn | UUGC Wind, PALM |

where $G$ is the geostrophic wind and $f_c$ the Coriolis parameter.

The two equation methodology explained above is used by Ellipsys3D, CFDWind and Alya. In the case of Meteodyn, a one equation RANS turbulence model $k$-$l_m$ is instead used (Delaunay, 2007). This methodology consists of solving the turbulence kinetic energy (TKE) equation, Eq. (4), replacing $\varepsilon$ in terms of $k$ and a parametrized mixing length $l_m$. Thus $\varepsilon = \varepsilon(k, l_m)$.

Furthermore, assuming the canopy elements exert a drag force on the flow, effects of the plant drag inside the canopy on the main flow are parametrized, presuming the form drag dominance, in the momentum Eq. (2) as:

$$\overline{f_i} = \overline{f_{D,i}} = -C_d a \,|\, \overline{\mathbf{u}} \,|\, \overline{u_i} \tag{12}$$

### 4.2.1 Meteodyn

**Model description**

Meteodyn WT is a commercial site assessment software that models the surface boundary layer (no Coriolis force included) using RANS, in particular a one-equation $k$-$l_m$ turbulence model and wall functions based on the Monin-Obukhov theory (Delaunay, 2007, 2013; Hurley, 1997). This turbulence closure scheme uses a prognostic equation for the turbulent kinetic energy and a mixing length approach for estimating the turbulent diffusion. Meteodyn WT version 5.2.1 was used in the present investigation.

The forest model in Meteodyn is based on a mean flow model, which treats the forest as a porous media (Costa, 2007), similar to what is used in other commercial solvers like WindSim (Crasto, 2006). A volumetric sink term is introduced in the momentum equations for all cells lying inside or partially covering the forest volume. The volumetric force depends on the drag coefficient $C_d$, which is a function of the forest density. In the Meteodyn simulations presented in this paper we used a value of $C_d^* = 0.005$.

In Meteodyn the canopy height $h$ for each cell is directly derived from the local roughness using the relation $h = C z_0$, where $C$ is an adjustable constant usually set to 20 and $z_0$ the roughness length at the cell surface. Cells with roughnesses higher than

0.8 cm are considered forest in the domain. Meteodyn can only handle roughness information provided in the .map format (contour lines information as in the WASP software (WASP)). For this reason, it was not possible to use the detailed PAD information provided to the modellers. The Meteodyn simulations presented here use roughness maps derived from SRTM data obtained using windPRO (WindPro).

Two versions of the forest model are available in Meteodyn, which differ in the computation of the mixing length in the one-equation turbulence model, named as robust and dissipative in the Meteodyn documentation. The dissipative forest model is used in the Meteodyn simulations presented in this investigation, where a 15 m extra high dissipation zone is used above the forest.

### Numerical setup

The computations are performed employing a cartesian structured mesh on a square domain with the dimensions of $13.5 \times 13.5$ km$^2$ and a 2.9 km height. The mesh is refined around the metmast location, with a grid stretching factor of $1.1$ in the horizontal and $1.2$ in the vertical directions. The final mesh has 224,000 cells. On the vertical direction the lines are always orthogonal to the topography surface.

Monin-Obukhov inlet profiles for velocity are defined at the inlet of the domain, as well as a constant turbulent kinetic energy

(Richards and Hoxey, 1993). The sides of the domain are defined as symmetry planes. The top and outlet sides of the domain are set with pressure outflow boundary conditions. At the domain surface wall functions are used, based on the local roughness of the cell and thermal stability classes.

### 4.2.2   EllipSys3D

### Model description

EllipSys3D is a CFD solver designed for various wind engineering applications – e.g. atmospheric boundary layer flows, turbine rotor computations, etc. It is a multi block finite volume solver of the incompressible Navier-Stokes equations in the general curvlinear coordinates. It uses collocated variable arrangement, employing the revised Rhie/Chow interpolation technique in order to avoid the odd-even pressure coupling. The pressure velocity coupling in the present study was based on the SIMPLE algorithm. Furthermore, the code is designed based on a non-overlapping domain decomposition technique, which combined

with its MPI parallelization, enables it to highly efficiently run on distributed/shared memory high performance computations (HPC) systems.

The standard and modified model constants according to Table 2 are used in the EllipSys3D setup in the present work. The geostrophic wind chosen is $G = 13$ m/s, giving the maximum length scale of $l_{max} = 28.71$ m. A 1D-precursor computation has been conducted in order to obtain the suitable inlet profiles, applied at all inlet boundaries.

To be able to model the effects of surface roughness on the mean flow and avoid resolving the laminar sub-layer, wall-functions as boundary conditions at wall surface boundaries are typically applied. In EllipSys3D, the wall boundary is placed

on the top of the roughness elements, this allows large near surface velocity gradients to be resolved using shallow (high aspect ratio) computational cells. The wall shear stress is accordingly used to specify the wall boundary conditions for momentum- and $\varepsilon$- equations, while a Neumann boundary condition is set for $k$ - , for a detailed description see Cavar et al. (2016) . A uniform roughness of 0.1 m is applied on the entire wall surface. The laser scan map provided for the present benchmark extending over a 52.5 km $\times$ 52.5 km zone centred at the Rygningsäs site location, in principle covering the whole wall surface area in the present study, was also fully incorporated into the EllipSys3D computations.

**Numerical setup**

The computational domain is a circular grid with a radius of 17 km, centred at the Rygningsnäs metmast location. The inner zone surrounding the site has a quadratic form. It is based on equally spaced grid points and covers a zone of 5 km $\times$ 5 km. The inner zone domain fully resolves the underlying topography, while the topography in the outer (buffer) zone is gradually smoothed towards the outer boundary. The same computational grid is used for all three investigated cases (flow directions), only the inflow and outflow boundaries on the grid circumference were adjusted for the each single run accordingly. Two grid sizes are considered, one using 512 $\times$ 512 grid points in the inner zone and 128 points in the outer (buffer) zone and the other one using 128 $\times$ 128 grid points in the inner zone and 64 points in the outer zone. The 3D grid was constructed by using an EllipSys3D default hyperbolic grid generator. 192 points were used in the vertical direction, with the first cell located at 1 cm above the terrain. The vertical hyperbolic mesh growth was controlled, so the zone up to a 50 m height had cells not stretching higher then 1 m. The top boundary was located at a 9 km height. The considered grids had approximately 100 million grid points (3072 blocks of $32^3$) in the 10 m resolution run and approximately 9.5 million grid points (288 blocks of $32^3$) in the 50 m resolution run.

### 4.2.3 Alya

**Model description**

Alya is a HPC code developed at the Barcelona Supercomputing Centre (BSC) to run large-scale applications. The code was recently tested on $100,000$ processors and showed a parallel efficiency above $90\%$ (Vazquez et al., 2016). The $k$-$\varepsilon$ model has been implemented and used in Alya for the present work.

The $k$-$\varepsilon$ model in Alya uses the modified model constants according to Table 2. Roughness-based wall functions are applied as boundary conditions at the ground to avoid solving the laminar sub-layer. A wall law satisfying the Monin-Obukhov equilibrium is imposed to the momentum and turbulence equations removing a boundary layer of thickness $\delta_w$. The wall shear stress is imposed on the momentum equation in terms of two velocity scales. Zero diffusion through the wall is imposed for the TKE, and the dissipation $\varepsilon$ is imposed in terms of the TKE value at a distance $\delta_w$ from the ground. For a detailed description see Avila et al. (2017).

The Navier-Stokes eq. (2) and turbulence equations (4)-(5) are discretized using a stabilised finite-element method using equal interpolation for all the unknowns. The Algebraical Subgrid Scale method (ASGS) was used (Codina, 1998), extended for nonlinear equations (Avila et al., 2015), to provide stability to transport and Coriolis dominating terms in the momentum equation, and to transport and production/dissipation terms in the turbulence equations, permitting the removal of spurious oscillations. The ASGS method also gives stability to pressure, allowing equal interpolation spaces for pressure and velocity. The velocity-pressure problem is decoupled using an Orthomin solver (Houzeaux et al., 2011) that converges to the monolithic scheme.

A robust finite element scheme written in block-triangular form (Codina and Soto, 1999) is obtained for the $k$-$\varepsilon$ equations (4)-(5). In order to avoid instabilities and numerical convergence issues, the $k$ and $\varepsilon$ unknowns are not allowed to drop below a predefined limit by applying a clipping. In addition, the innermost iterative loops of the $k$ and $\varepsilon$ equations (4)-(5) are linearized using a Newton-Raphson scheme for the quadratic terms, considering $\nu_t$ and $P_k$ constants within the innermost loops.

Once the algebraical system of equations are obtained, a Deflated Conjugate Gradient (Lohner et al., 2011) solver with a linelet pre-conditioner (Soto et al., 2003) is used to solve the pressure, and a Generalized Minimizing Residual (GMRES) solver is used for the velocity and turbulence unknowns.

## Numerical setup

The Ryningsnäs problem was solved using a cylindrical mesh with a radius of 20 km. The mesh is centred on the metmast. Surrounding the metmast the mesh resolution is 10 m over a $4\,\text{km} \times 4\,\text{km}$ horizontal square. Farther from the mast the horizontal mesh size grows until it reaches 500 m horizontal element length. The vertical resolution starts with a 0.5 m element length close to the wall and being of 1.2 meters inside the forest. The computational domain has a vertical extension of 2000 m.

The inflow boundary conditions are defined from a precursor simulation over flat and homogeneous terrain (i.e. single column model 1D). The obtained fields are used also as initial conditions. Zero traction is imposed over the outflow boundaries. No velocity penetration and zero tangential stress are imposed over the top boundary.

Three different geostrophic velocities were set to the three different wind directions to match the desired velocity at mast. The geostrophic velocities were set to 12.7 m/s, 13.2 m/s and 12.7 m/s for the wind directions of 100, 240 and 290 degrees respectively.

### 4.2.4 CFDWind

#### Model description

CFDWind is a modelling framework developed at CENER on top of the open-source CFD platform OpenFOAM (Weller et al., 1998) version 2.4.0 (ope, 2015). The model is designed for the simulation of atmospheric boundary layer flows through the solution of the incompressible RANS equations in which turbulence closure is achieved by the eddy-viscosity theory and a modified version of the $k - \varepsilon$ closure scheme Apsley and Castro (1997) described in the previous section.

As only neutral atmospheric stability was considered, the flow is assumed stationary so the SIMPLE algorithm is employed to solve the pressure-velocity coupling while 2nd-order upwind schemes are used for the discretization of both velocity and turbulence convective terms.

The Coriolis apparent force is added explicitly to the momentum equation together with the horizontal pressure gradient that drives the system which is derived from the hydrostatic relation for stationary cases.

The perturbations induced by forests are modelled by adding drag and source/sink terms in the momentum and turbulence-closure equations, respectively, as proposed by Sogachev and Panferov (2006). Table 2 (Modified2 $k-\varepsilon$) shows the drag and closure-model constants employed for the simulations. Rather than tuned, these values follow the set employed by Detering and Etling (1985) which are derived from the experiments carried out in Panofsky et al. (1977).

Despite that it is expected that wind flow will be dominated by the effects of forest features near the surface, $z_0$-based wall functions are implemented as boundary conditions at the ground assuming wall bounded flow. That is, the applied horizontal kinematic shear stress is set via an effective eddy-viscosity $\nu_t^{wall}$ which together with the dissipation rate and production term of the turbulent kinetic energy equation are obtained with the local velocity scale $u_*^{wall}$ computed from values of velocity and turbulent kinetic energy of the cells adjacent to the ground (see Chávez-Arroyo et al. (2014) for more details).

Similar to EllipSys3D, the wall functions consider that the computational grid is placed on top of the roughness elements so that restrictions related to the height of the cells adjacent to the ground and $z_0$ are avoided and high aspect ratio cells can be used. Outlet conditions are specified at the sides and at slip (only tangencial velocity and no-gradient) conditions are prescribed at top of the domain.

**Numerical setup**

The numerical grid was created with the meshing software WindMesh. The tool has been developed jointly by BSC and CENER for the automatic and fast generation of grids over terrain. There are currently two different versions further developed by each institution: BSC-WindMesh (Gargallo-Peiró et al., 2015) which was employed in the simulations of the Alya model, and the CENER-WindMesh (Gancarski and Chávez-Arroyo, 2017) version which was used for the generation of the grids for the CFDWind and UUCG runs.

CENER-WindMesh creates structured terrain-following grids optimizing parameters such as orthogonality and skewness by applying filters to the 2D (ground) mesh, and elliptic smoothing techniques for the final 3D mesh. The mesh is designed so that terrain is smoothed far from the area of interest whereas towards the central zone the cells are refined to the maximum resolution established. Only real topography is considered for the grid generation in the center. The "transition" zone between boundaries and the central zone is a progressive blend between the real terrain and flat boundaries.

Similar to previous approaches, a precursor run is conducted prior to the full-terrain simulation (successor) in order to create the equilibrium profiles that serve as inlet conditions. Precursor simulations are conducted on flat domains with periodic boundary conditions on the sides with the top and wall-treatment mentioned above. The PAD is set to a constant value of

19 $\mathrm{m}^2\mathrm{m}^{-3}$ with values of roughness length and forest height of 0.72 m and 14 m, respectively. For the successor runs, the heterogeneous roughness values are created based on the canopy height map $H$, using the simple relation of $z_0 = H/20$.

The value of the geostrophic wind is chosen so that the velocity magnitude obtained in the simulations is approximately 10 $\mathrm{ms}^{-1}$ at 100 m at the position of the mast for each of the three flow cases. The values are 14.66, 15 and 15 $\mathrm{ms}^{-1}$, resulting

in maximum length scales of $l_{max} = 32.2$, $l_{max} = 33$ and $l_{max} = 33$ m for the inflow directions of 100, 240 and 290 degrees respectively.

The computational domain is square-shaped and covers an extension of 18 km $\times$18 km $\times$3 km centered on the Ryningsnäs tower. From that, only a 12 km $\times$12 km region considers real topography in which PAD and $z_0$ data are interpolated from the input canopy information. The rest of the domain is set as a flat, buffer area with the same PAD, $z_0$ and $H$ of the precursor

simulation. For each flow case, the mesh is rotated in order to align the wind direction with the normal vector of the inlet patch at 100 m above ground. The meshes consist of $20 \times 10^6$ cells with 60 vertical levels. The first cell height is set to 1 m and then grows with a geometric function with a constant growth rate of 1.08.

## 4.3   LES

### 4.3.1   UUCG-Wind

**Model description**

The computations by UUCG were carried out with an solver implementation based on the OpenFOAM platform, version 3.0.1. A neutrally-stable wind flow is computed with LES coupled with a SGS model (Yoshizawa and Horiuti, 1985; Yoshizawa, 1986) where $\nu_{\mathrm{SGS}}$ is estimated from the subgrid turbulence kinetic energy $k_{\mathrm{SGS}}$ which is in turn computed from a transport equation.

It is assumed that the forest acts as a porous surface exerting a drag on the flow. This is represented in the simulation with the introduction of a source term in the LES momentum equation (eq. 2):

$$f_{D,i} = -C_D a \left| \langle \boldsymbol{u} \rangle \right| \langle u_i \rangle \tag{13}$$

where $C_D$ is the forest drag coefficient, $a$ is the frontal-area-density (assumed here to be equal to the PAD). This approach has been successfully used in wind computations over forests with LES, e.g. by Nebenführ (2015) and Boudreault (2015). The

employed value of $C_D = 0.2$ throughout the domain is taken from the latter. While the effect of the forest in eq. (13) is applied over the resolved part of the velocity field, the dissipative effect of TKE caused by the forest is included within the subgrid scales by adding the term

$$\varepsilon_{\mathrm{SGS}} = -\frac{8}{3} C_D a \left| \langle \boldsymbol{u} \rangle \right| k \tag{14}$$

to the transport equation of $k_{\mathrm{SGS}}$.

A wall model is also used to account for the roughness of the ground, although it is expected that its influence on the wind flow will be much smaller in comparison to the forest. For this, the wall model implementation available in the OpenFOAM

libraries of SOWFA (Churchfield et al., 2014) was employed. The velocity deficit due to the interaction with the ground is introduced indirectly, by means of applying a surface stress. For this, the model of Schumann (1975) is used, where the non-zero components of the stress tensor at the surface are computed as a function of the friction velocity, which in turn is calculated from the logarithmic law with a local time-average for the horizontal velocity. Only the modules corresponding to the modelling of the surface stress are used from SOWFA, importing these from OpenFOAM 2.x into the version used for the simulations.

**Numerical setup**

The computational domain consists of a box with the dimensions $32 \, \mathrm{km} \times 20 \, \mathrm{km} \times \sim 1.2 \, \mathrm{km}$ in the longitudinal, spanwise and vertical directions. (the height varies due to the differences in terrain elevation for each wind direction). The met mast is located at $20 \, \mathrm{km}$ in the longitudinal direction, in the mid-spanwise crossing. The mesh is created using CENER-WindMesh, described in section 4.2.4, producing a mesh with a varying ground elevation. In this way, the grid consists of zones in the horizontal plane: a farm zone ($20 \, \mathrm{km} \times 12 \, \mathrm{km}$, metmast at $14 \, \mathrm{km}$) at the interior which is then successively surrounded by a transition zone and a buffer zone. The two outermost regions can be described as rectangular edges with a width in the longitudinal and spanwise directions of 3 km and 2.5 km for the transition zone and 3 km and 1.5 km for the buffer zone. The arrangement of the grid in the horizontal plane is uniform in the farm and buffer regions, while the cells stretch in the transition zone, changing their size from that of the farm to that of the buffer zones. The terrain becomes flat at the buffer edges until, at the outermost boundary (with a width of 500 m), the elevation is equal in all sides with a value of 63.06 m (100 degrees), 163.25 m (240 degrees) and 137.76 m (290 degrees). The horizontal cell resolution is 25 m for the farm and 250 m in the buffer regions. The height of the first cell at the location of the metmast is approximately 3.4 m while the size increases vertically with a growth rate of $\sim 1.05$. The domain height as well as the number of cells in the vertical direction in every case is $1.172 \, \mathrm{km}$ and $84$ cells for 100 degrees, $1.305 \, \mathrm{km}$ and $86$ cells for 240 degrees as well as $1.267 \, \mathrm{km}$ and $85$ cells for 290 degrees.

The longitudinal axis of the domain is aligned with the wind direction for each case, so the inlet is perpendicular to the inflow. All lateral boundaries are set to periodic boundary conditions. Hence, the inlet flow is recycled from the outlet. The flow is driven by a uniform pressure gradient, following the procedure described by Bechmann (2006) which also comprises the introduction of Coriolis forcing (assuming a latitude of 57 degrees). In this manner, the pressure gradient is calculated for the desired geostrophic wind, which is set as to yield the desired wind velocity at 98 m for each case. The complete height of ABL is simulated to avoid the parametrization of the components of the shear stress, as they become negligible at this altitude. The ground surface is set to a wall with a uniform roughness of $z_0 = 0.03$ m, while the PAD for the cells covering the tree area is extracted (by linear interpolation) from the file at $10 \, \mathrm{m} \times 10 \, \mathrm{m}$ resolution, using the same method as for CFDWind in section 4.2.4. For each wind direction, simulations are run during about $400 \times 10^3$ seconds to develop the flow and achieve convergence of second-order statistics. Results are obtained from values averaged during subsequent computations lasting $20 \times 10^3$ s with $\Delta t = 0.296$ s, yielding a maximum Courant-Friedrichs-Lewy number of CFL $\approx 0.6$ over the whole domain.

### 4.3.2 PALM

**Model description**

PALM is a massively parallelized LES solver designed for studies of the atmospheric and oceanic boundary layer. It is an open source code (PAL) and has been applied to the simulation of a variety of atmospheric boundary layer studies in the past 20 years. PALM solves the filtered, incompressible, non-hydrostatic Navier-Stokes equations under the Boussinesq approximation on an equidistant Cartesian grid. The sub-grid scale turbulence is parameterized by a 1.5th order closure after Deardorff (1980) solving a prognostic equation for $k_{\mathrm{SGS}}$. Dirichlet (no-slip) boundary conditions are prescribed at the surface. Further details on the numerics and physics of PALM can be found in Maronga et al. (2015).

The forest effect is modelled by adding a sink term to the momentum equation following Shaw and Schumann (1992) and Watanabe (2004). Furthermore a sink term is added to the prognostic equation for the $k_{\mathrm{SGS}}$ according to Shaw and Schumann (1992) to ensure a rapid breakdown of turbulence in the canopy. A source term is added to the temperature equation allowing to prescribe a heat flux at the canopy top to account for the effect of incoming solar radiation. See Kanani et al. (2014) for equations and further details of the canopy model. A forest canopy can be prescribed by specifying the tree height and a vertical PAD profile. The PAD profile can be prescribed by using a beta probability density function (parameters $\alpha$, $\beta$ and leaf area index) or by specifying PAD values at discrete levels and doing a piecewise linear reconstruction. The latter approach has been used for the benchmark simulations. As per default, only a single PAD profile and tree height can be specified and hence, only a homogeneous forest can be simulated. Simulating a heterogeneous forest would have required significant code development which has not been feasible in the scope of the benchmark.

**Numerical setup**

The benchmark simulations use a model domain of 2304 m $\times$ 1152 m $\times$ 1867 m with a grid spacing of 4 m. A homogeneous forest canopy is prescribed by setting periodic horizontal boundary conditions and using averaged PAD profiles where each of the three sectors has been averaged over the innermost 2 km. A conventionally neutral atmospheric boundary is simulated by prescribing a constant potential temperature up to a height of 500 m capped by a stable layer with a gradient of 1 K / 100 m. Coriolis force is considered assuming a latitude of 57 degrees North. The roughness length is set to 0.1 m. A geostrophic wind speed of $u_g$ = 13.0 ms$^{-1}$ and $v_g$ = -9.3 ms$^{-1}$ had to be prescribed to achieve a mean wind along the $x-$ direction at 100 m height of about 7.4 ms$^{-1}$ as demanded by the benchmark specification. The simulations have been run for 10 h to reach a steady state. The results have been averaged over the entire horizontal model domain and over the last 30 min of the simulation. Additionally, a vertical point profile at the centre of the domain, averaged over 2 h, has been provided.

### 4.4 Numerical set up overview

To summarise, four different RANS codes and two different LES codes are included in the study. Forest modelling is basically done in the same way in all codes apart from Meteodyn. All models apart form PALM uses heterogeneous forest, but Meteodyn

is based on a different surface data set. Domain sizes are similar apart from PALM which uses a significantly smaller domain, but since PALM has homogeneous forest with re-circulation the domain size is directly comparable. A summary of some key modelling properties is found in Table 3. The numerical challenge stretches from the use of a commercial code to state-of-the-art research codes using up to about 40 million cells, modelling of 20 000 physical seconds and the use of approximately 20 000 CPU hours.

**Table 3.** Numerical setup. Cell size refers to the horizontal grid size in the inner domain.

| Model | Cell Size [m] | $a \times b \times c$ [km] | $G$[m/s] | Mesh generator |
|---|---|---|---|---|
| Meteodyn | 15 | $13.5 \times 13.5 \times 2.9$ | | |
| EllipSys3D hi res 1 | 9.8 | 34 (diameter) $\times 9$ | 13 | |
| EllipSys3D hi res 2 | 9.8 | 34 (diameter) $\times 9$ | 13 | |
| EllipSys3D low res 1 | 39.1 | 34 (diameter) $\times 9$ | 13 | |
| EllipSys3D low res 2 | 39.1 | 34 (diameter) $\times 9$ | 13 | |
| Alya | 10 | $20 \times 20 \times 2$ | 12.7, 13.2, 17.7 | WindMeshGargallo-Peiró et al. (2015) |
| OpenFoam CFDWind | 12 | $18 \times 18 \times 3$ | 14.6,15,15 | WindMesh Gancarski and Chávez-Arroyo (2017) |
| OpenFOAM UUCG | 25 | $32 \times 20 \times 1.2$ | 11.5, 12.5, 11.4 | WindMesh Gancarski and Chávez-Arroyo (2017) |
| PALM | 4 | $2.3 \times 1.2 \times 1.9$ | 16.0 | |

## 5 Results

One main purpose of RANS and LES modelling within the wind energy community is to extrapolate tower measurements vertically and spatially. In the next section, the vertical extrapolation (vertical profiles) is reported first, followed by the horizontal extrapolation (planes).

### 5.1 Vertical profiles

Wind speed, wind veer and turbulence are crucial to power production. These three quantities, in the form of mean wind speed $S = \sqrt{U^2 + V^2}$, mean wind direction profile and turbulent kinetic energy TKE $= 0.5(\overline{u'^2} + \overline{v'^2} + \overline{w'^2})$ are evaluated in Fig. 3, Fig. 5 and Fig. 6 and respectively.

Modelled and measured profiles are shown for the three different wind directions described in Section 3.2. As is apparent from studying Fig. 3 most models actually show a slightly lower wind speed than the targeted 7.4 m/s at 100 m height. The wind profiles are also provided in logarithmic height coordinates and it is apparent that the measurements have a deeper log-linear region than most of the modelled curves.

Most models overestimate the wind speed gradient, reported in Fig. 4. The overestimation increases with height and in the upper layers is close to a factor of two. The fact that most models have a lower 100 m wind speed compared to the target 7.4 m/s means that one has to be careful when interpreting how good the models are at estimating the wind speed gradient,

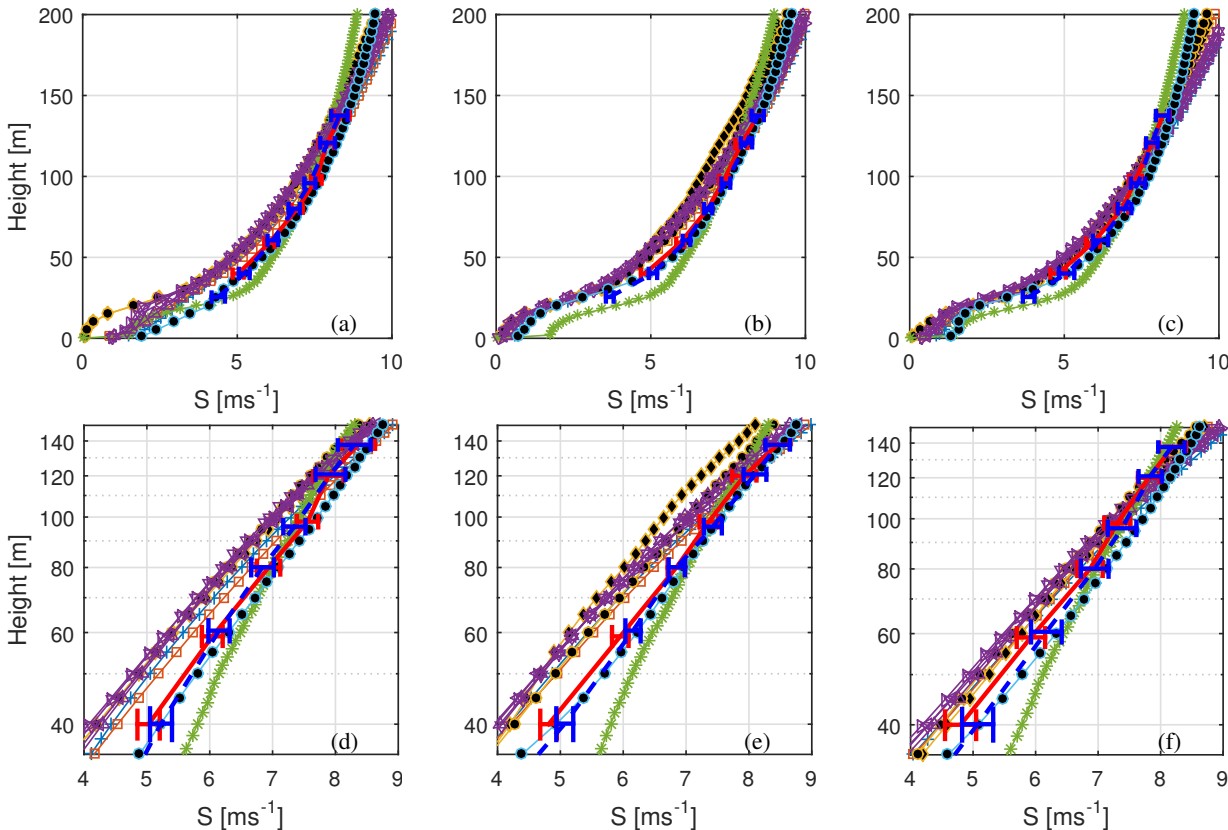

**Figure 3.** Wind speed profiles.The blue dashed line shows the average from the cups, the red full line shows the average from the sonics. Error bars indicate the 95 % confidence level for the mean value. The various other markers indicate simulated wind speeds. Please see table 1 for a legend of marker representations. LES models have been given black filling in order to increase readability. (a) and (d) shows results from 100 °, (b) and (e) from 240 ° and (c) and (f) from 290 °.

but a 0.5 m/s difference in 100 m wind speed roughly means a $5 \times 10^{-3}$ s$^{-1}$ difference in wind speed gradient (assuming the difference is spread out over 100 m). As can be seen in Fig. 4 the difference, for most models, is at least twice as large and is consistently over predicted by all RANS models except Meteodyn. Meteodyn is the only RANS not to use the PAD input and thus it is difficult to know whether the smaller shear is due to the surface boundary condition or some other modelling aspect.

5 In terms of shear, one can see that OpenFoam UUCG stands out as the best performing model. The forest parameterization and PAD data were the same for that model as for the RANS models using PAD, so the difference in shear cannot be explained by difference in amount and placement of the surface drag elements. The main difference between OpenFoam CFDwind and OpenFoam UUCG is that the latter is run in LES mode which seem to result in either better estimation of the boundary layer height or more realistic mixing of the velocity deficiencies, or both. The other LES model, PALM, did not run with the detailed

10 PAD input, but did use averaged PAD profiles averaged with the innermost 2 km radius for each of the three directions. As seen in Fig. 3 and 4 PALM does have lower shear compared to the RANS models, but it is unclear how much of that is due to the

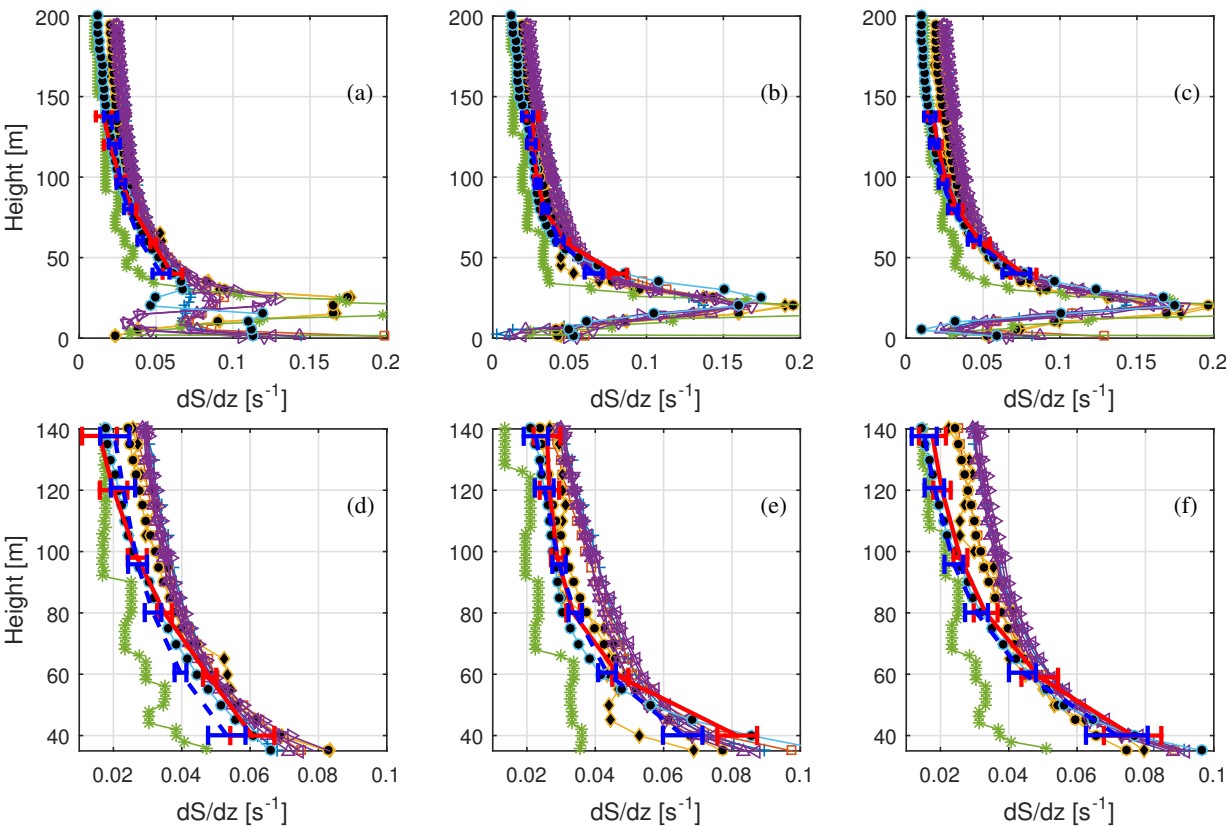

**Figure 4.** Modelled and observed wind shear. The lower three plots are zoomed in on the instrument heights to increase readability. The blue dashed line shows the average from the cups, the red full line shows the average from the sonics. Error bars indicate the 95 % confidence level for the mean value. The various other markers indicate simulated wind speeds. Please see table 1 for the marker representations. LES models have been given black filling in order to increase readability. (a) and (d) shows results from 100 °, (b) and (e) from 240 ° and (c) and (f) from 290 °. The upper row and the lower row show the same results, but the lower is zoomed in and is in log-space to facilitate better readability.

LES effect and how much is due to the constant PAD profile used. One interesting thing to notice is that while all the models using the detailed PAD fields, as well as the measurements, they have the largest shear at 100 m in the 240° sector, PALM shows the largest shear in 290° at 100 m despite having much lower PAD in that direction. This is in line with analytical theory for homogeneous forests which predict a maximum of the roughness length at moderate PAD after which blocking effects lead to gradually lower roughness lengths with increasing plant area index (PAI) Jackson (1981). Most forests considered for wind energy have a substantial heterogeneity, though, and as can be seen in Fig. 1b the lower average PAD within the innermost 10 km does not come from areas with less dense forest, but rather from a large area *without* forest, over which the flow may be able to adopt to a lower roughness environment.

In an earlier publication (Arnqvist et al., 2015) it was shown that the wind turning with height (veer) was considerable at rotor heights, especially in stable stratification. Most models participating in the study show a veer of 1-3 degrees between 50 and 150 m (Fig. 5), this is about half of the veer found by the measurements, but the kinking of the measured curves also indicate the difficulty in measuring small deviations of the wind direction as wind load on the tower and booms as well as alignment accuracy all add to the uncertainty. The general shape of the wind direction profile in the Ekman layer however suggests that the models represent the relevant physics accurately apart from the Meteodyn model where wind direction strangely does not seem to be coupled to the balance between Coriolis force, stress divergence and pressure gradient. Another very interesting point is that all the models using detailed PAD input show a reverse in the wind direction turning centered between 20 and 60 m, placed lowest in 240° and highest in 100°. This reverse could be an indication that the main driver of the flow is turbulence transport from above, and given the Ekman spiral aloft, the flow carries momentum directed to the right hand side, leading to an counter clockwise turning with height within the forest. This behaviour has recently been shown to govern the direction of the wake behind a wind farm (van der Laan and Sørensen, 2017). It should however be noted that earlier field studies have shown a prevalence of clockwise turning with height also within the forest (Smith et al., 1972; Pinker and Holland, 1988). Unfortunately the measurement site lacked wind measurements within the forest, and the mechanism behind the forest wind direction profile has not been further investigated in this study.

Out of the three directions, 290° stands out as the one with the lowest overall mean PAD, as seen in Fig. 2. The main contributor to the lower PAD average is however mainly low forest or lack of forest in the far upstream region which is apparent from Fig. 1 (b) where an area without forest is seen at around 7-12 km from the metmast in the fetch in the 290° direction. It is likely it is this low roughness area that causes the TKE level to drop more quickly with height in 290° direction compared with the other directions (Fig. 6). While all models seem to get the overall level of TKE approximately right only the LES version of OpenFoam captures the decrease of TKE with height for all three directions. The RANS models show much less variation between the different directions. Interestingly, the PALM LES shows almost no difference in TKE between the different directions even though the average PAD is different in all three directions, see Fig. 2 ($d$). It should be noted though, that the wind speed at 100 m in PALM was lower in 240° direction which may explain why the turbulence level is not higher for that sector.

The purple lines in Fig. 6 shows the different setups of EllipSys3D RANS, and while the effect of resolution does not seem to influence the results much (used values are found in Table 3), the choice of turbulence closure constants play a huge role

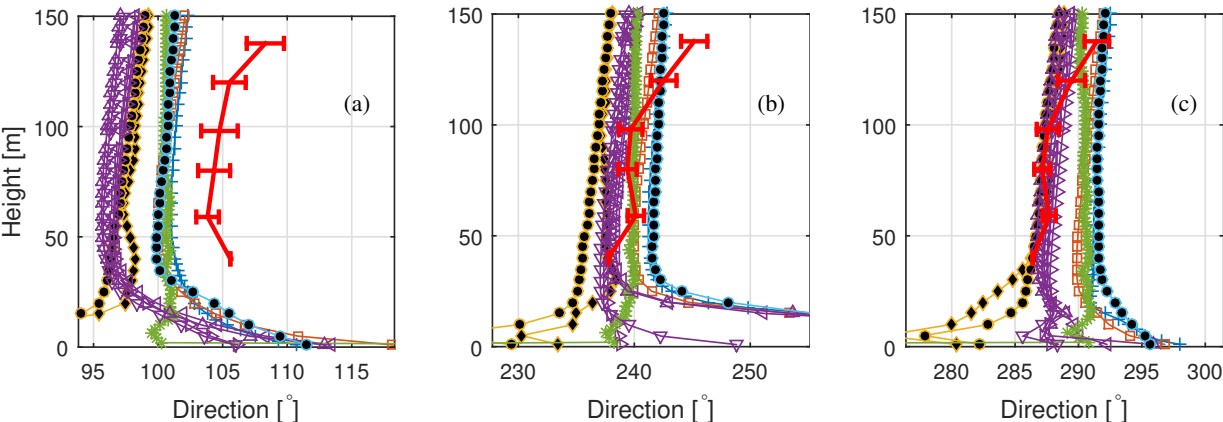

**Figure 5.** Wind direction profiles. The red full line shows the average from the sonics. Error bars indicate the 95 % confidence level for the difference between the wind direction at each height and direction at 40 m. The various other markers indicate simulated wind directions. Please see table 1 for the legend of marker representations. LES models have been given black filling in order to increase readability. (a) shows results from 100 °, (b) from 240 ° and (c) from 290 °.

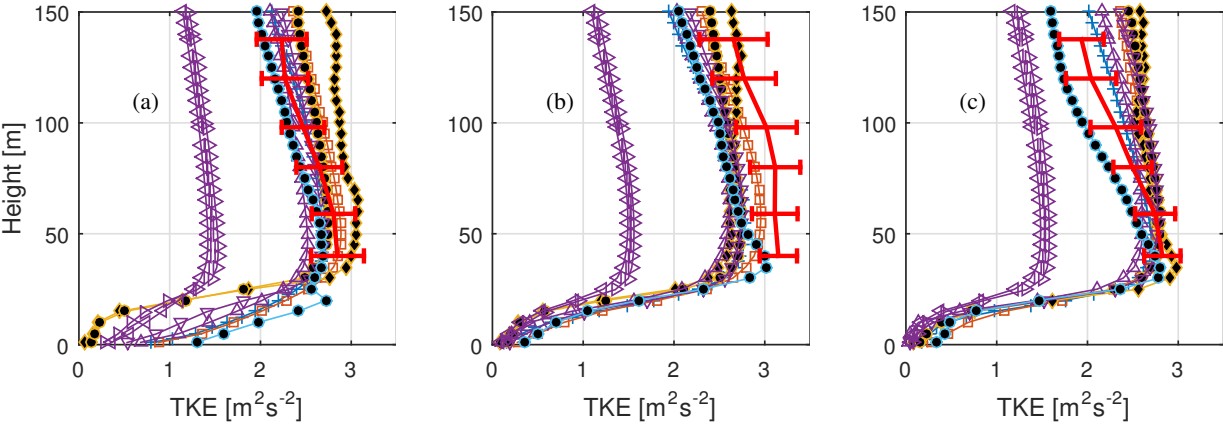

**Figure 6.** TKE profiles. The red full line shows the average from the sonics. Error bars indicate the 95 % confidence level for the mean value. The various other markers indicate simulated TKE levels. Please see table 1 for a legend of the marker representations. LES models have been given black filling in order to increase readability. (a) shows results from 100 °, (b) from 240 ° and (c) from 290 °.

(values found in Table 2). The use of standard $k - \varepsilon$ values produce only about half of the TKE compared to the values tuned for atmospheric boundary layers despite the shear being virtually the same.

## 5.2 Horizontal planes

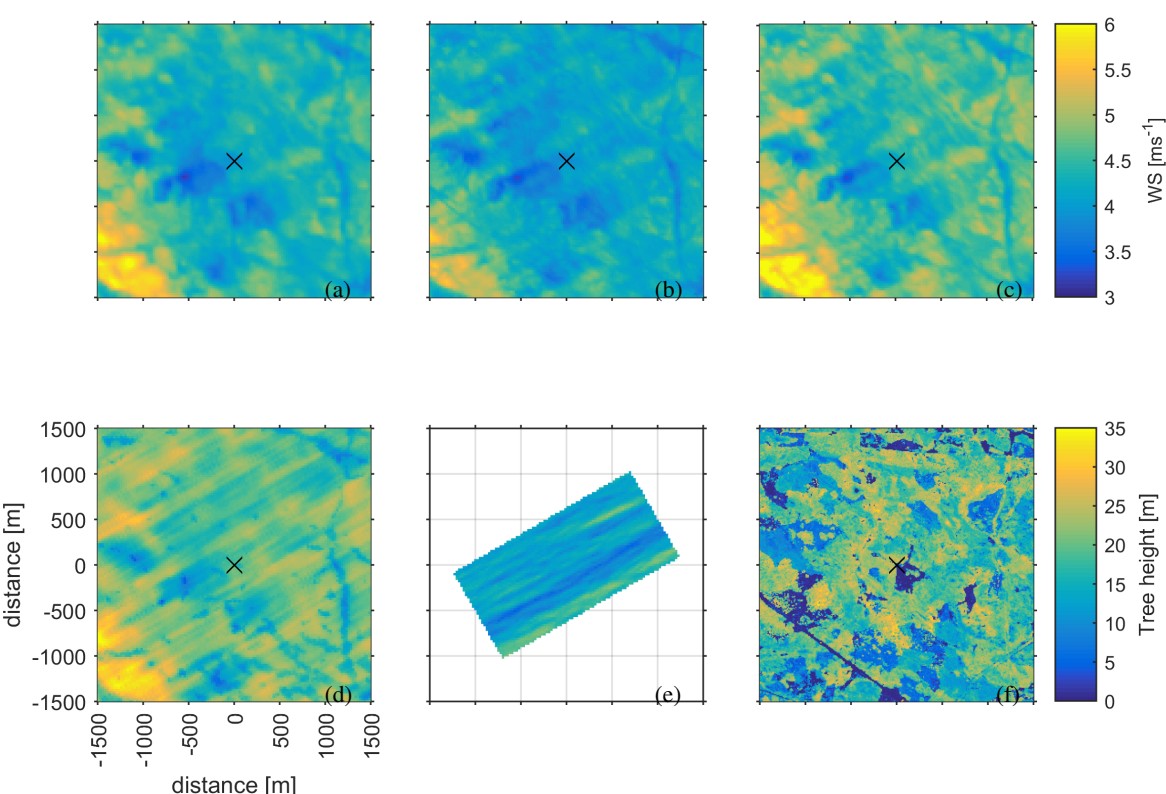

**Figure 7.** (a)-(e) Simulated wind speed at 40 m above the local ground height. (a) EllipSys3D hi res 1, (b) Alya, (c) OpenFoam CFDWind, (d) OpenFoam UUCG and (e) PALM. The size of the boxes is 3 km x 3 km centered around the measurement tower (marked by an x). (f) The forest height.

In order to evaluate spatial differences the modellers were instructed to submit horizontal planes surrounding the measure-
5  ment tower. Planes are shown here for 40 m above the local elevation, Fig. 7, and 140 m above the local elevation, Fig. 8. Also displayed in the figures are tree height and terrain elevation. Although there is some correlation with tree height at 40 m, most of the correlation is with elevation, reflecting that the height is above local terrain and all models can be seen to have wind fields with streamlines that are smoother than the terrain, which results in higher wind speed over high terrain and lower wind speed over low terrain. This feature is common for all models dealing with varying terrain.
10  Although the models all show similar wind speed patterns there is a difference in the amount of wind speed streaks present and in the strength of the streaks. All the models show more intense streaks at the higher height. The LES models show more

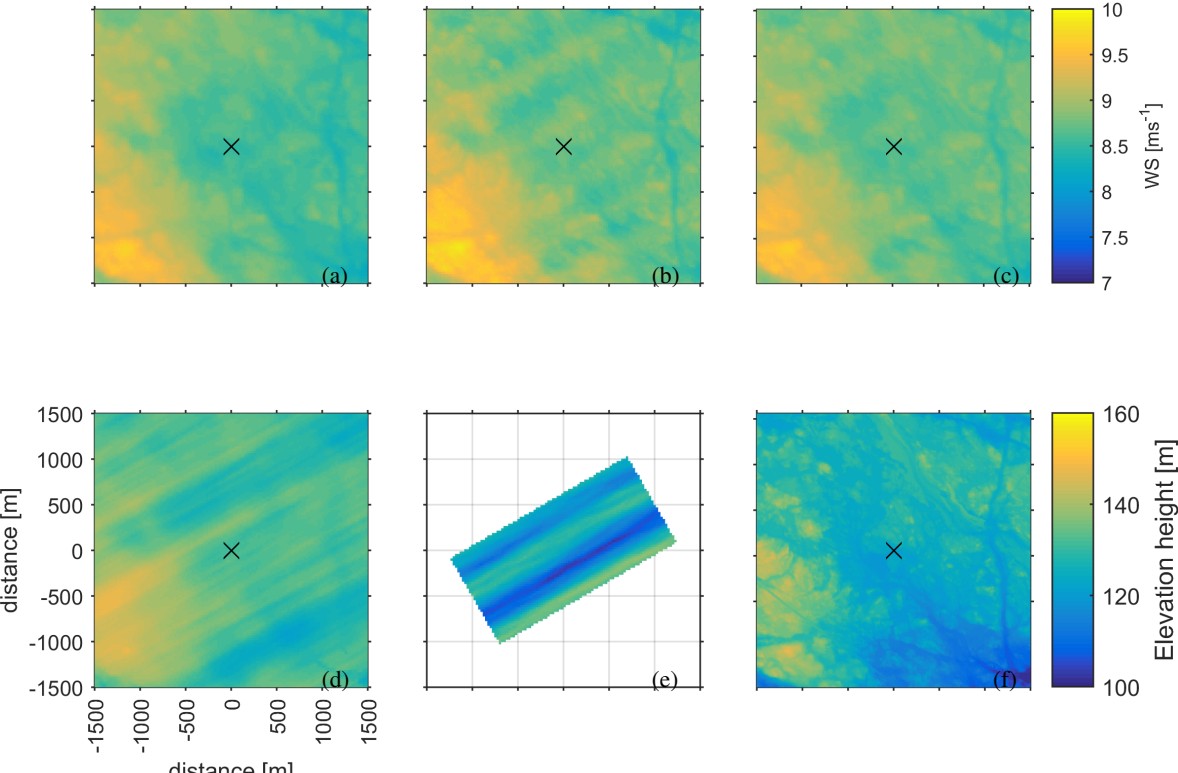

**Figure 8.** (a)-(e) Simulated wind speed at 140 m above the local ground height. (a) EllipSys3D hi res 1, (b) Alya, (c) OpenFoam CFDwind, (d) OpenFoam UUCG and (e) PALM The size of the boxes is 3 km x 3 km centered around the measurement tower (marked by an x). (f) The elevation height.

tendency for streaks than the RANS models. EllipSys3D shows almost no streaks whereas Alaya and OpenFoam CFDWind have similar streak patterns as the LES OpenFoam UUCG. The streaks correlate with topographical features, but there are also clear streaks in the PALM LES which ran without topography.

## 6 Discussion and Conclusions

The main aim of the study has been to investigate the state of the art abilities of modelling groups to replicate measurements in neutral conditions over a forested site. Six modelling groups in total completed the whole process and submitted results. The RANS modellers using research codes used a fairly homogeneous approach to the model task, while the LES modellers took quite different approaches. Overall a variety of options were used and in this section we will try to discuss some of the implications of these different choices.

## 6.1 Use of surface data

All models except Meteodyn and PALM use ALS input for topography and forest data. The fact that a variety of models (including LES and RANS) were able to use the ALS input was considered a success. The use of PAD data from ALS removes the uncertainty of having to guess the PAD or the roughness length and displacement height, which in practice can be a large source of uncertainty when estimating the wind resource at a potential site. The only model not to use the ALS was Meteodyn (instead deriving PAD in some form) and that model also clearly stands out in the validation. Yet, some of the differences may be also attributed to the use of first order turbulence closure.

An interesting observation noted by several of the participants was that the roughness was totally dominated by the drag of the forest and that the value of the ground roughness did not make a visible impact on the results. That could on the other hand be expected, since even though the forest characteristic is heterogeneous, the landscape as a whole can be considered forest covered.

## 6.2 Foot print

An initial question at the start of the study was whether the differences in the measured profiles between the three directions would be captured by the models, given the detailed surface data. In summary the differences between the directions turned out to be small for the RANS models, but the LES model that used the detailed surface data produced profiles that resembled the measured profiles.

From Fig. 4 it is clear that the majority of models overpredict the shear and one may be led to believe that the forest representation, ALS conversion to PAD, is causing this discrepancy, but OpenFoam LES does in fact match the shear very well using the same forest data and the same $C_d$ value. Especially interesting is the difference between OpenFoam LES and RANS, which are computed with implementations based on the same platform and the same grid generator. The reason for this discrepancy is an interesting point for further study.

The LES version of OpenFoam furthermore showed a much more pronounced difference between the inflow angles, both in terms of shear and TKE, a possible explanation may be that the RANS models are overdiffusive, something also indicated by the fact that RANS models show less streaks in the horizontal planes.

## 6.3 Closure constants

One of the most striking outcomes of the study is that the $k-\varepsilon$ closure with standard constants produces far too little TKE. The difference is attributed to the value of $C_\mu$, where a value of around 0.03 seems to give reasonable TKE level. The conclusion of the participating modellers is that constants following Sogachev and Panferov (2006) should be used for future studies. Also worth mentioning is the point that all of the RANS models (apart from Meteodyn) show a too high shear, in fact almost by a factor two in the upper layers, and therefore they would be expected to generate higher levels of TKE than found in the measurements. This is also the case in the 290° direction (Fig. 6 (c)), but not in the 100° direction where the shear also is too high in the upper parts.

## 6.4 Resolution and domain size

As seen in Table 3, the two modelling attempts that had the lowest resolution were Ellipsys 3D low res and OpenFoam LES at 39.1 m and 25 m respectively. This does however not seem to affect the results in a negative way. The difference between the low res and high res Ellipsys is very minor, both in comparison to the measurements and in comparison to the difference of changing the closure constants. The relatively low resolution OpenFoam LES captured the wind profile well for all heights and all directions, and it seems to be important to accurately model the PAD and topography in the footprint. In order to accurately represent the turbulence moments the LES resolution needs to be such that the resolved part of the flow is large enough. For the OpenFoam LES the ratio of $k_{\mathrm{SGS}}$ to the total $k$ was around 20 % at the tree tops and 10 % at 100 m, and while $k$ was rather well predicted, and the spatial resolution is probably on the limit of accurately representing the individual components of $k$, at least in the lower part of the boundary layer.

Following the reasoning in Wyngaard (2010) page 224, the upstream domain size needs to be on the order of $zU/u_*$ to capture inhomogeneities affecting the wind at height $z$, which for the current site is around 15 km, exactly the upstream distance for the inner domain of the OpenFoam LES simulations. An interesting point for future studies is to examine how the results depend on upstream domain size for flow modelling with heterogeneous surface data.

Another computational expense in LES modelling is the integration time. This particular study was aimed at simulating a stationary case, and since the Coriolis force may introduce inertial oscillations it is important to make sure that the influence of oscillations does not impact the results. Another important conclusion of the study is that stationary, neutral conditions are very rare in the atmosphere, and hence future studies should investigate naturally occurring transient conditions such as diurnal cycles, evening transitions and developing unstable turbulence.

The orders of magnitude difference in numerical challange, both in the set up and in used computational time, should be considered when considering the accuracy of the modelling results.

## 6.5 Recommendation for future studies

Many modellers expressed the difficulties involved in trying to determine the correct value of the $U_g$ or pressure gradient in order to match the target 100 m wind speed. While it would be possible to instead normalize the results with, $u_*$ or some other appropriate quantity, the fact that the upwind topography and PAD seem crucial for good results point to the fact that normalization needs to be done with care since different wind speeds and turbulence levels would imply differences in the fetch, especially in stratified conditions. Furthermore the use of a turbulence scaling parameter introduces substantial uncertainty to the measurements since the statistical uncertainty is much larger for second order moments than first order moments.

Based on the problem of adjusting forcing in order to match a target wind speed, measurement campaigns designed for flow model validation should attempt to measure the boundary conditions and forcing, such as the boundary layer height, vertical and horizontal fluxes, radiation, ground temperature and geostrophic wind speed. Another option is to use a MESO-scale model to compute the boundary forcing for the micro-scale models, but then care has to be taken that no additional uncertainty is introduced due to bias between the MESO-scale model results and reality.

Future microscale model comparisons at complex forested sites should focus on the modelling of thermal stability effects. The radiative cooling during cold nights strongly affects the wind profile, introducing a strong variation of heat flux in the forest. Heat transfer models over forested sites have been already implemented over flat terrain. However, new thermal models need to be developed and validated accounting for complex terrain. It is also clear that the given thermally stratified conditions it is more difficult to compare stationary conditions, both due to model drift and to the fact that the atmosphere is in itself mostly in transition. Comparing natural cycles would also remove the uncertainty of integration time since the physical time modelled would be the same for all participants. The representativeness of the results would also increase significantly if thermally stratified conditions were included since the strictly neutral conditions used in this study are rare in the atmosphere. Such a study would of course necessitate the use and development of unsteady RANS models.

*Data availability.* The data used to validate the models (selected as described in section 3) as well as the PAD values are available upon request.

*Author contributions.* Stefan Ivanell coordinated the project. Johan Arnqvist selected the data, was responsible for the ALS to PAD conversion, gathered the results and made the plots. Matias Avila ran Alya, Dalibor Cavar ran EllipSys3D, Roberto A. Chavez-Arroyo ran CFDWind, Hugo Olivares-Espinosa ran UUCGWind, Carlos Peralta and Jamal Adib ran Meteodyn and Björn Witha ran PALM. All authors contributed to writing the text.

*Acknowledgements.* The work is mainly performed within the ERANET+ project NEWA. This work was partly conducted within StandUp for Wind, a part of the StandUp for Energy strategic research framework in Sweden. The simulations were partly performed on resources provided by the Swedish National Infrastructure for Computing (SNIC) within the project SNIC 2017/11-10. Ebba Dellwik is greatly acknowledged for contributions regarding benchmark design, converting the ALS to PAD and measurement data selection.

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
