# Peer review of "Microscale model comparison (benchmark) at the moderate complex forested site Ryningsnäs"

_Wind Energy Science, 2018_

## Referee Comment (RC1) · Anonymous Referee #1 · 16 May 2018

Review of the manuscript WES-2018-20

Microscale model comparison (benchmark) at the moderate complex forested site Ryningsnäs

By Ivanell et al.

Summary: This paper reports on a benchmark study in which a series of numerical models for wind energy purposes is tested for a forested site in the north of Sweden. Several codes are tested for a cases under neutral conditions, and permutations are being made for closure constants. It appears that the models somewhat fit the observed profiles, although a few don't. In more detail the modelled wind shear and TKE differ substantially from the observations for some of the models. Model intercomparisons can be very powerful efforts to learn about differences in model behavior (and not so much as a competition who is the best). The difficulty here is that the models that took part are so different in forcing and domain and grid settings, that it is difficult to draw general conclusions easily. There are a number of points that the paper can be improved upon, which has been listed below.

Recommendations: major revision needed

Major remarks:

Abstract: I think the abstract should be rewritten, it is a bit short and not so informative on what you really learnt. E.g. from the statement that the turbulent closure constants have a huge impact not much is learnt. Try to come with real recommendations.

Benchmark description: Coming from the meteorological community I have some difficulties with understanding the forcings. First the term target wind is unclear. In my view the wind at 100 m is a result of all external forcings which are the pressure gradient, advection and surface friction. So one should better defend the choice to go for a target wind. Personally prescribing a geowind would be more feasible and fair. Secondly it is unclear to me for which day and time the wind profile is forecast. Do we look at a single time slot, or averages over multiple days? In additional the lead time of the forecast is never mentioned. Obviously models needs some spin up, too short spin up may result in poor scores. At the same time relatively long forecasts with a long lead time are expected to be more off from the observation (predictive skill horizon). In addition I was missing whether the models also apply a temperature budget equation that allow for katabatic winds/slope effects since the slope is substantial. Finally, the paper should mention whether advection of momentum in u and v was applied or not.

Discussion: Overall I have the feeling the discussion section of the paper can be improved. Hence a more in depth discussion about the quality of the used observations, a more elaborated discussion of the model results with respect of the model results already reported elsewhere in the literature (i.e. do models behave the same as in
the benchmark study). Moreover, also the representativeness of the case has to be improved since it was (only 1 day?) for neutral conditions? Finally, what is the next step in case a new intercomparison should be developed, what should we focus on, a case with more stability, or a case with more/less complex terrain/vegetation? For the scientific progress in the future, some more lights could be shed on these aspects.

Remarks:

P1, ln 2: unclear whether you performed tuning here or whether it is a free forecast. I think readers from the meteorological community have difficulty with understanding the target wind, since the wind at 100 m should be a results of the forcings and the dynamics.

P1, ln 12; reword: new areas. You mean new sites, not new techniques or so.

P1, ln 16: elaborate on wind and turbulence: what is the problem/are the challenges with these variables. You probably mean that wind shear and turbulence should be limited for wind turbines, for load assessment etc..

P2, ln 17: add a comma behind as such

P2, ln 18: coming benchmarks: please be more concrete if you are aware of current initiatives.

P2, end of section 1: please add a few lines that model intercomparison studies have been very fruitful methods to improve the model scores by adjusting parameterizations after the confrontations with other model results and observations, and add a few examples, e.g.

Bosveld, F.C., P. Baas, Gert-Jan Steeneveld, Albert A.M. Holtslag, Wayne M. Angevine, Eric Bazile, Evert I.F. de Bruijn, Daniel Deacu, John M. Edwards, Michael Ek, Vincent E. Larson, Jonathan E. Pleim, Matthias Raschendorfer, Gunilla Svensson, 2014: The GABLS third intercomparison case for model evaluation, Part B: SCM model intercomparison and evaluation, Bound.-Layer Meteor., 152, 157-187.

P3., ln 19: make clear whether the 138 m is above ground level or canopy top level. Idem for P4, ln 6.

P6, ln 3: please correct the definition of u*, a square is missing in the vw part

P6, ln 5: please mention which percentage of the observations fulfilled the criteria. As such we can learn something about the representativeness of the case.

P6, ln 13: Why the experiment wasn't opened for non NEWA colleagues? Perhaps new people could have been entrained in the community

P6, ln 21: notation: above you use the overbar for time averages and here you switch to <>. Please make consistent. Idem for P7 ln 4 were the overbar suddenly denotes the resolved scales of the LES.

P6, table 1: explain how the PALM LES can run in column mode? Is it thereby a RANS model?

P7, section 4.2: here it is a bit unclear to me whether all models use the same descriptions that are presented and general for all of them, especially eq 9 and 11. Equation 11 is also just a parameterization, though not much in use anymore in meteorology since it appeared not to be very useful. Idem for table 2: did all models use these coefficients?

P9 and further: structure. I think it is better for the readibility to make a section 4.2.1 Meteodyn and then 4.2.1a for model description and 4.2.1b for numerical setup. And then for all models.

P9, table 2: elaborate the table caption. Unclear which model uses which closure in this way. Also defend why the combinations of exactly those values were taken. There seems to be no strategy, or the strategy is not presented in the paper.

P10, ln 18: please defend the 13 m/s that was set. Earlier I learnt that the geowind should be predicted and should be compared in the model outcome. Also what was the

source for the 13 m/s? Was it taken from ERA-Interim or so or from a radio sounding? Please defend.

P17, figure 3: caption: Modelled and observed wind speed profiles...... Please add day and time of the forecast, as well as the lead time. Also help the reader by adding that def are zoom in plots of abc.

P17, ln 7: please correct the overbars.

P17, ln 8: order of referencing of figure confusing.

P18, figure 4, caption: Modelled and observed wind shear

P18, ln 6: overestimate (plural)

P18, ln 8: superscript the 1 of ms-1

P19, ln 18: As can be seen: avoid passive sentence.

P19, ln 19: half of the veer; please refer here to the literature. It is a common problem that NWP models underestimate the turning of the wind with height.

P19, ln 26-27: I am not convinced pressure gradient is not important, since it is the major driver of all flows. Though I agree its relative impact will be smaller than aloft. Please reword or quantify all terms of the momentum equation.

Figure 6: The family of runs that has a low TKE value has a TKE value that appears to be exactly a factor 2 smaller than the others. Some of those model report 2*TKE as the prognostic variable. Could that explain the difference, or perhaps different modellers applying the same model but one of the two did not divide by 2. It is just a suggestion.

Figure 8: The results of this figure should be discussed in more detail. So far I can see different values but the reason behind them remain unclear. Perhaps also better to plot them as histograms of the values over the domain so we can better see the mean value and percentile differences. Also: what time of the day is this?

P23, ln 4: please unravel the causes behind the differences in more detail.

P23, ln 12: too little

---

## Referee Comment (RC2) · Anonymous Referee #2 · 3 Jul 2018

Review of "Microscale model comparison (benchmark) at the moderate complex forested site Ryningsnas" by Ivanell et al.

The model intercomparison study is poorly designed; in addition, the manuscript is quite poorly written. It reads as if different sections were written by different authors; the authors should have made a sincere effort to produce a "homogenized" manuscript.

1.The authors failed to understand that geostrophic wind is not a "tuning" parameter. It represents the balance between the synoptic-scale pressure gradient force and Coriolis force. The coordinators of this intercomparison study should have fixed the geostrophic wind values.

2.In addition, different participants used different roughness values. This is also not

acceptable in a model intercomparison study.

3.Furthermore, integration times of different models vary significantly. For example, the UUCG-Wind model was run for 400,000 seconds. Whereas, the PALM model was run for 10 h (36,000 seconds).

When Coriolis term is involved, the models produce inertial oscillation with a period of 2*pi/f. So, all the models should have been run for a fixed time-period in order to have an "apples-to-apples" comparison.

4.PALM model did not simulate a "truly" neutral boundary layer. Rather, they imposed a capping inversion and simulated a "conventionally" neutral boundary layer. Other models did not incorporate such a strategy.

5.Page 2, line 23: The authors wrote: "Using PAD data instead of estimated roughness lengths may be a way to reduce the uncertainties of site assessment..." However, in the intercomparison study different groups prescribed different roughness lengths. One participant did not use PAD data at all. The others used this dataset in a variety of ways. Again, an intercomparison study should not be conducted in this manner.

6.All the wall functions used by different models should have been properly described.

7.Caption of Figure 2: the subplot (d) is not described.

8.Page 6, definition of ustar (below Eq. 1) has a typo.

9.Page 6, line 18: "simplified version": a Reynolds-averaged (RANS) or Filtered (LES) version of the N-S equation is not a "simplified" version.

10.Page 6, line 20: "time averaged" should be replaced by "ensemble averaged".

11.Page 7, line 2: "fully resolved" -> atmospheric LES runs are rarely fully resolved. One has to demonstrate that everywhere in the domain, the subgrid-scale energy is less than 20% of total energy.

12. Table 1: PALM uses Deardorff's closure. This is a $k_{SGS}$ closure.

13. Page 8, line 16: "limiting maximum length" -> change to "limiting mixing length" or "asymptotic mixing length".

14. Page 11, line 13: "et al (2016)" -> missing author.

15. Page 11, line 18: What is a "reactive term" in turbulence equations?

---

## Author Comment (AC1) · 5 Oct 2018

Dear Reviewer,

Thank you for you constructive comments.

In the following we address the comments made, and point out the changes in the revised manuscript. Parts that have been rewritten or added have been highlighted in green in the revised version of the manuscript. General edits, and edits that are common to both referee suggestions have been made in red.

Best regards Stefan Ivanell with co-authors.

Please also note the supplement to this comment:

[Figure]

https://www.wind-energ-sci-discuss.net/wes-2018-20/wes-2018-20-AC1-supplement.pdf

[Figure]

**Supplement:**

**Answers to referee 1**

Stefan Ivanell[1], Johan Arnqvist[1], Matias Avila[2], Dalibor Cavar[3], Roberto Aurelio Chavez-Arroyo[4], Hugo Olivares-Espinosa[1], Carlos Peralta[5], Jamal Adib[5], and Björn Witha[6]

[1]Uppsala University, Wind Energy Section, Campus Gotland, 621 67 Visby, Sweden
[2]Barcelona Supercomputing Center, BSC, Spain
[3]Wind Energy Department, Technical University of Denmark, Denmark
[4]National Renewable Energy Centre (CENER), Spain
[5]Wobben Research and Development MS GmbH, Germany
[6]ForWind - Carl von Ossietzky Universität Oldenburg, Germany

*Correspondence to:* Stefan Ivanell (stefan.ivanell@geo.uu.se)

We appreciate the constructive comments, which we have taken into consideration in the revised version. In the following we address the comments made, and point out the changes in the revised manuscript. Parts that have been rewritten or added due to comments by the referees have been highlighted in green in the revised version of the manuscript. General edits, and edits that are common to both referee suggestions have been made in red. Smaller edits, such as spelling and grammatical corrections are noted below, in the specific comments, but are not highlighted by colour in the manuscript

**1   General comments**

We appreciate the time the referee has taken into providing examples of how the manuscript can be improved. We agree with the reviewer that the text could be improved in many places.

> 1. *Abstract: I think the abstract should be rewritten, it is a bit short and not so informative on what you really learnt. E.g. from the statement that the turbulent closure constants have a huge impact not much is learnt. Try to come with real recommendations.*

We have rewritten the abstract in order to better reflect what we perceive as the main conclusions of the study, in an as concrete way as possible.

> 2. *Benchmark description: Coming from the meteorological community I have some difficulties with understanding the forcings. First the term target wind is unclear. In my view the wind at 100 m is a result of all external forcings which are the pressure gradient, advection and surface friction. So one should better defend the choice to go for a target wind. Personally prescribing a geowind would be more feasible and fair. Secondly it is unclear to me for which day and time the wind profile is forecast. Do we look at a single time slot, or averages over multiple days? In additional the lead time of the forecast is never mentioned. Obviously models needs some spin up, too short spin up may result in poor scores. At the same time relatively long forecasts with a long lead time are expected to be*

*more off from the observation (predictive skill horizon). In addition I was missing whether the models also apply*
*a temperature budget equation that allow for katabatic winds/slope effects since the slope is substantial. Finally,*
*the paper should mention whether advection of momentum in u and v was applied or not.*

Thank you for the comment. The choice of having a fixed 100 m wind speed has been raised as concern also by the other

5 reviewer. It is also our view that the 100 m is the result of forcing rather than a forcing in itself. The idea behind the seemingly awkward choice of having a target wind speed is twofold. First, in wind energy assessment it is customary to tune the model results (in some way) to the measured wind speed at the given site, hence the idea follow a tradition within the field where micro scale models are used to extrapolate the measured wind conditions spatially. Second, it is the idea of the study to see how results of different models, run with best practice, compare to the measured wind and turbulence profile at the site. Since

10 the geostrophic wind speed was not measured during the measurement campaign there is simply no possibility to have that as a known forcing. One could naturally set a value for the geostrophic wind speed and compare scaled values instead. That does however introduce a number of problems. The boundary layer height is connected to the geostrophic wind speed, and we did not measure that either. Having a different geostrophic wind speed in the measured wind profiles as compared to the modelled wind profile would also introduce the uncertainty that the foot print would be different. Since the surface conditions are highly

15 heterogeneous, our intention was for the models to have as similar foot print as possible to the actual (real) foot print. Finally, if scaled values were compared, one would not know whether it was the turbulence that was ill predicted or the wind speed, in case the profile does not match. Put it in another way, an initial question of the study was: Given a specific 100 m wind speed, does the model compute a realistic turbulence level given highly realistic surface boundary conditions?

Please see the benchmark description section in the manuscript for the updated text regarding this issue.

20 The measurement data was selected through strict selection criteria out of around one year of total measurement time. The process was described under the measurement section, but we have added small parts in to make it clearer.

Since the models are micro scale flow models, the results **should not be** viewed upon as forecasts, but rather as a stationary result of the forcing and boundary condition. Since the target period is stationary, no advection terms were used, and in fact all of the inputs given to the modellers have been described in the first paragraph of the section Benchmark description.

25 The models participating in the benchmark are all except for PALM strictly neutral, meaning that they do not solve a temperature equation. PALM was instead run without temperature forcing. The respective information is in each model description.

It is an interesting point that you bring up on the possibility of modelling katabatic flow, however we disagree about the significance of the slope at the particular site. The overall slope is only around 90 m in 50 km, so the site can be considered flat. We have brought up the question in the discussion as interesting for future studies though.

30 3. *Discussion: Overall I have the feeling the discussion section of the paper can be improved. Hence a more in depth discussion about the quality of the used observations, a more elaborated discussion of the model results with respect of the model results already reported elsewhere in the literature (i.e. do models behave the same as in the benchmark study). Moreover, also the representativeness of the case has to be improved since it was (only 1 day?) for neutral conditions? Finally, what is the next step in case a new intercomparison should be developed,*

*what should we focus on, a case with more stability, or a case with more/less complex terrain/vegetation? For the scientific progress in the future, some more lights could be shed on these aspects.*

Thank you for the suggestions, we have rewritten the discussion and conclusions accordingly, and structured it so that the message becomes clearer.

**2   Specific comments**

1. *P1, ln 2: unclear whether you performed tuning here or whether it is a free forecast. I think readers from the meteorological community have difficulty with understanding the target wind, since the wind at 100 m should be a results of the forcings and the dynamics.*

We expect readers from the wind energy community to be familiar with this type of prediction, but thank the reviewer for pointing out the it may not be so clear to other readers. We have updated the abstract to better reflect the aim of modelling for general condition, rather than a specific forecast.

2. *P1, ln 12; reword: new areas. You mean new sites, not new techniques or so.*

Well, both are true, but the intent was on sites. We have clarified the text.

3. *P1, ln 16: elaborate on wind and turbulence: what is the problem/are the challenges with these variables. You probably mean that wind shear and turbulence should be limited for wind turbines, for load assessment etc..*

Furthered the explanation.

4. *P2, ln 17: add a comma behind as such*

Added

5. *P2, ln 18: coming benchmarks: please be more concrete if you are aware of current initiatives.*

Clarified to microscale benchmarks. There are new benchmarks in the pipeline in NEWA, but citing is difficult. We added a citation where the experiments and their purpose is explained.

6. *P2, end of section 1: please add a few lines that model intercomparison studies have been very fruitful methods to improve the model scores by adjusting parameterizations after the confrontations with other model results and observations, and add a few ex- amples, e.g. Bosveld, F.C., P. Baas, Gert-Jan Steeneveld, Albert A.M. Holtslag, Wayne M. Angevine, Eric Bazile, Evert I.F. de Bruijn, Daniel Deacu, John M. Edwards, Michael Ek, Vincent E. Larson, Jonathan E. Pleim, Matthias Raschendorfer, Gunilla Svensson, 2014: The GABLS third intercomparison case for model evaluation, Part B: SCM model intercom- parison and evaluation, Bound.-Layer Meteor., 152, 157-187*

Added

> 7. *P3., ln 19: make clear whether the 138 m is above ground level or canopy top level. Idem for P4, ln 6.*

Clarified it's above ground.

> 8. *P6, ln 3: please correct the definition of u\*, a square is missing in the vw part*

Corrected.

> 9. *P6, ln 5: please mention which percentage of the observations fulfilled the criteria. As such we can learn something about the representativeness of the case.*

Added.

> 10. *P6, ln 13: Why the experiment wasn't opened for non NEWA colleagues? Perhaps new people could have been entrained in the community*

The benchmark was not closed for outside participation, but the call for participation was limited. In the end only NEWA members participated, but to avoid confusion we removed the sentence. Future NEWA benchmarks are advertised in a broader way. Please visit thewindvaneblog.com if you want to learn more.

> 11. *P6, ln 21: notation: above you use the overbar for time averages and here you switch to <>. Please make consistent. Idem for P7 ln 4 were the overbar suddenly denotes the resolved scales of the LES.*

Sorry for that, we have now consistently used the overline to represent time average and <> for the LES filter.

> 12. *P6, table 1: explain how the PALM LES can run in column mode? Is it thereby a RANS model?*

The two PALM references in table 1 are both full 3D LES, actually it is the same run but two different data sets are extracted. "Spatial average" means that the output was horizontally averaged in each height. "Vertical column" means that data from a virtual met mast in the center of the model domain was extracted.

> 13. *P7, section 4.2: here it is a bit unclear to me whether all models use the same descriptions that are presented and general for all of them, especially eq 9 and 11. Equation 11 is also just a parameterization, though not much in use anymore in meteorology since it appeared not to be very useful. Idem for table 2: did all models use these coefficients?*

We have added a sentence to make the text clearer. The individual model descriptions now refer to Table 2 for the constants that they were using. A proper formulation for the length scale limiter is interesting and an interesting point for future research, especially with unsteady RANS simulations.

14. *P9 and further: structure. I think it is better for the readibility to make a section 4.2.1 Meteodyn and then 4.2.1a for model description and 4.2.1b for numerical setup. And then for all models.*

Good suggestion. Unfortunately the style does not permit subsubsub sections, but we will bring this suggestion to the editor.

15. *P9, table 2: elaborate the table caption. Unclear which model uses which closure in this way. Also defend why the combinations of exactly those values were taken. There seems to be no strategy, or the strategy is not presented in the paper.*

It is correct that there was no strategy. The participants were given the instructions to use their best practice, which means that there was no prior control of the choice of closure constants. We have updated the caption and added a new line to show which constants were used by which group. We also furthered the discussion on closure constants in the discussion and conclusion part to better highlight the importance of the $C_\mu$ value.

16. *P10, ln 18: please defend the 13 m/s that was set. Earlier I learnt that the geowind should be predicted and should be compared in the model outcome. Also what was the source for the 13 m/s? Was it taken from ERA-Interim or so or from a radio sounding? Please defend.*

Please see the comment on the general question.

17. *P17, figure 3: caption: Modelled and observed wind speed profiles...... Please add day and time of the forecast, as well as the lead time. Also help the reader by adding that def are zoom in plots of abc.*

The predicted wind speeds are not a forecast as such, but should approximate the measured wind speed in neutral stationary flow. We have updated the introduction and abstract to make this fact clearer. We have updated the caption according to the suggestion.

18. *P17, ln 7: please correct the overbars.*

Corrected.

19. *P17, ln 8: order of referencing of figure confusing.*

Changed the order to the order they appear.

20. *P18, figure 4, caption: Modelled and observed wind shear*

Updated according to the suggestion.

21. *P18, ln 6: overestimate (plural)*

Corrected

22. *P18, ln 8: superscript the 1 of ms-1*

Corrected

23. *P19, ln 18: As can be seen: avoid passive sentence.*

We have reformulated the sentence in order to be more direct to the point.

24. *P19, ln 19: half of the veer; please refer here to the literature. It is a common problem that NWP models underestimate the turning of the wind with height.*

Since this study only include micro-scale models, the difference in model structure from the usual NWP model is fairly large. Even though the mechanism behind the underestimation of the veer may be the same for both model types, it seems slightly speculative to assume so since the NWP model would also take into account baroclinicity and evolving boundary layers, but would not include the forest drag explicitly. We are not aware of a good reference that thoroughly investigates the matter, but if the reviewer has a good suggestion we would certainly consider including it.

25. *P19, ln 26-27: I am not convinced pressure gradient is not important, since it is the major driver of all flows. Though I agree its relative impact will be smaller than aloft. Please reword or quantify all terms of the momentum equation.*

Thank you for bringing up this is interesting point. We agree with the reviewer that the original sentence was speculative without any reference to support the argument. In the literature (Smith and Carson 1972, Pinker and Holland 1988), wind direction change within the forest has mostly been attributed to a balance between pressure gradient and flux divergence, arguing that Coriolis force is small due to the low wind speeds. Those studies has also observed a clockwise change with height, in contrast to our study. The anti clockwise turning in our study would therefore suggest that pressure gradient becomes smaller in the balance, perhaps due to a dominance of turbulence transport. This counter intuitive behaviour has also been reported from studies of wake propagation behind wind turbines (Van der Laan and Sørensen 2017). We have updated the manuscript with the arguments and references above. A more detailed study would be interesting but is out of the cope of this study.

26. *Figure 6: The family of runs that has a low TKE value has a TKE value that appears to be exactly a factor 2 smaller than the others. Some of those model report 2\*TKE as the prognostic variable. Could that explain the difference, or perhaps different modellers applying the same model but one of the two did not divide by 2. It is just a suggestion.*

That was the initial observation made by the authors as well, but the reason behind the large difference is almost entirely made up by the difference in the $C_\mu$ constant, being either 0.09 (corespodning to low TKE) or 0.033 (giving high TKE). This is an important conclusion and we hope that it comes through with clarity in the updated manuscript.

*27. Figure 8: The results of this figure should be discussed in more detail. So far I can see different values but the reason behind them remain unclear. Perhaps also better to plot them as histograms of the values over the domain so we can better see the mean value and percentile differences. Also: what time of the day is this?*

The main reason behind including the planes is to illustrate the difference in flow structure between the models. For example streakiness. We have chosen to omit a more quantitative analysis mainly for two reasons. First, we have no validation data, and a pure model to model comparison should (as both reviewers have pointed out) be done in a more controlled fashion, and second, the open call of the benchmark means that the models do not run on the same grid, hence a comparison of point to point prediction would have to involve the additional complication of interpolating to a common grid, which should be fine for qualitative analysis, but should perhaps be done with care for quantitative analysis. We have therefore chosen to keep the discussion of the planes limited.

The snapshots are not from a specific time of the day, but should rather represent all stationary, neutral conditions. Please see earlier comments.

*28. P23, ln 4: please unravel the causes behind the differences in more detail.*

We have updated the sentence to clearly point out the main difference, that is, Meteodyn is using a first order TKE closure.

*29. P23, ln 12: too little*

Corrected.

The comments from the reviewer have certainly helped us to improve our manuscript and we hope that the comments have been taken into consideration satisfactorily.

[revised manuscript text omitted]

---

## Author Comment (AC2) · 5 Oct 2018

Dear Reviewer,

Thank you for you constructive comments.

In the following we address the comments made, and point out the changes in the revised manuscript. Parts that have been rewritten or added have been highlighted in blue in the revised version of the manuscript. General edits, and edits that are common to both referee suggestions have been made in red.

Best regards Stefan Ivanell with co-authors.

Please also note the supplement to this comment:

[Figure]

https://www.wind-energ-sci-discuss.net/wes-2018-20/wes-2018-20-AC2-supplement.pdf

**Supplement:**

**Answers to referee 2**

Stefan Ivanell[1], Johan Arnqvist[1], Matias Avila[2], Dalibor Cavar[3], Roberto Aurelio Chavez-Arroyo[4], Hugo Olivares-Espinosa[1], Carlos Peralta[5], Jamal Adib[5], and Björn Witha[6]

[1]Uppsala University, Wind Energy Section, Campus Gotland, 621 67 Visby, Sweden
[2]Barcelona Supercomputing Center, BSC, Spain
[3]Wind Energy Department, Technical University of Denmark, Denmark
[4]National Renewable Energy Centre (CENER), Spain
[5]Wobben Research and Development MS GmbH, Germany
[6]ForWind - Carl von Ossietzky Universität Oldenburg, Germany

*Correspondence to:* Stefan Ivanell (stefan.ivanell@geo.uu.se)

We appreciate the constructive comments, which we have taken into consideration in the revised version. In the following we address the comments made, and point out the changes in the revised manuscript. Parts that have been rewritten or added due to comments by the referees have been highlighted in blue in the revised version of the manuscript. General edits, and edits that are common to both referee suggestions have been made in red. Smaller edits, such as spelling and grammatical corrections

5 are noted below, in the specific comments, but are not highlighted by colour in the manuscript

**1 General comments**

We appreciate the time the referee has taken into providing examples of how the manuscript can be improved. We agree with the reviewer that the text could be improved in many places.

10     1. *The model intercomparison study is poorly designed; in addition, the manuscript is quite poorly written. It reads as if different sections were written by different authors; the authors should have made a sincere effort to produce a "homogenized" manuscript.*

We agree that an ideal inter comparison study should be designed in a different fashion. However, this is not a pure model to model inter comparison study, but rather a best practice study to survey the different approaches used in the modelling community. We have updated the text throughout to more clearly reflect that aim; investigating how wind energy assessment micro

15 scale models compare to measurements given a typical problem. The model task is very typical of wind resource estimation, simulating the wind profile given a target 100 m wind speed knowing the topography and surface characteristics.

As we have rewritten the manuscript we hope that the new updated version is more homogeneous and has a more consistent style.

**2    Specific comments**

1. *The authors failed to understand that geostrophic wind is not a "tuning" parameter. It represents the balance between the synoptic-scale pressure gradient force and Coriolis force. The coordinators of this intercomparison study should have fixed the geostrophic wind values.*

Thank you for the comment. Whether or not the geostrophic wind speed is something that should, or should not be, adjusted depends on the reference point. From a modelling point of view it is natural that the geostrophic wind is a fixed forcing known, or set, a priori. From a measurement perspective it is quite natural that the geostrophic wind speed is to a large degree unknown, or in the rare cases when it is measured, connected with much larger uncertainty than, lets say, the 100 m wind speed. The choice of having a fixed 100 m wind speed has however been raised as concern also by the other reviewer, and a number of participants of the study has expressed the impracticality of having to adjust the geostrophic wind speed. The idea behind the seemingly awkward choice of having a target wind speed is twofold. First, in wind energy assessment it is customary to tune the model results (in some way) to the measured wind speed at the given site, hence the idea follow a tradition within the field where micro scale models are used to extrapolate the measured wind conditions spatially. Second, it is the idea of the study to see how results of different models, run with best practice, compare to the measured wind and turbulence profile at the site. Since the geostrophic wind speed was not measured during the measurement campaign there is simply no possibility to have that as a known forcing. One could naturally set a value for the geostrophic wind speed and compare scaled values instead. That does however introduce a number of problems. The boundary layer height is connected to the geostrophic wind speed, and we did not measure that either. Having a different geostrophic wind speed in the measured wind profiles as compared to the modelled wind profile would also introduce the uncertainty that the foot print would be different. Since the surface conditions are highly heterogeneous, our intention was for the models to have as similar foot print as possible to the actual (real) foot print. Finally, if scaled values were compared, one would not know whether it was the turbulence that was ill predicted or the wind speed, in case the profile does not match. Put it in another way, an initial question of the study was: Given a specific 100 m wind speed, does the model compute a realistic turbulence level given highly realistic surface boundary conditions?

Please see the benchmark description section in the manuscript for the updated text regarding this issue.

2. *In addition, different participants used different roughness values. This is also not acceptable in a model inter-comparison study.*

The aim of the study is not to compare models, but rather to compare model and best practice use. It is the authors opinion that changing the settings of a particular model may lead to larger differences than between different models, and a pure model comparison study would not answer the main question: Given current best practice, how well can the measured wind profile be matched by different modelling groups?

It should be noted that the surface conditions are very heterogeneous (in fact, all the 10x10 m grid cells are unique). Therefore, as long as the domain size, model cell size (vertically and horizontally) and drag parameterization are not exactly the same, the resulting roughness would not be the same anyways. We do not regard this as a problem, as much of the initial

questions of the study relates to how different use of the surface data leads to success of the task of matching the measured wind profile.

We have updated the text in the introduction to better reflect the aim of the study.

    3. *Furthermore, integration times of different models vary significantly. For example, the UUCG-Wind model was run for 400,000 seconds. Whereas, the PALM model was run for 10 h (36,000 seconds). When Coriolis term is involved, the models produce inertial oscillation with a period of 2\*pi/f. So, all the models should have been run for a fixed time-period in order to have an "apples-to-apples" comparison.*

This question relates only to the two LES models participating, since the RANS models were run until stationary (given each modellers/models definition of stationary). The existence of oscillations also largely depends on the initial balance at the start of the run, boundary conditions, filters etc, so it is difficult to draw conclusions on such a specific and detailed topic. Reflecting on the aim of the study to investigate models and modelling practices we do not see a problem with the different integration times as that is a very important part of running a LES model and integration time is part of the expenses that the modeller needs to handle. In other words, the study is not about comparing apples to apples, but rather to look at the whole fruit basket. In fact, the difference in integration time is a result of the study and we thank the reviewer for noticing the difference. We have updated the discussion section with a paragraph on integration time and oscillations.

    4. *PALM model did not simulate a "truly" neutral boundary layer. Rather, they imposed a capping inversion and simulated a "conventionally" neutral boundary layer. Other models did not incorporate such a strategy.*

See the answer to the above questions. In short, it was the intention that the modelling groups should use what they considered their best practice.

    5. *Page 2, line 23: The authors wrote: "Using PAD data instead of estimated roughness lengths may be a way to reduce the uncertainties of site assessment..." However, in the intercomparison study different groups prescribed different roughness lengths. One participant did not use PAD data at all. The others used this dataset in a variety of ways. Again, an intercomparison study should not be conducted in this manner.*

Please see the answer to the above questions. In addition to that, sensitivity tests done by the participants (not included in the publication) indicated no noticeable effect of the surface roughness. In other words, the total roughness/resistance is totally dominated by the drag of the forest. We have updated the discussion with a paragraph on the subject.

    6. *All the wall functions used by different models should have been properly described.*

The space is limited and since the flow in this case is dominated by the drag of the forest, not the drag of the surface, we have omitted that. We have added the domination of the forest drag to the conclusion and discussion section.

    7. *Caption of Figure 2: the subplot (d) is not described.*

Sorry, a (d) got mixed up with a (c). It is now corrected.

8. *Page 6, definition of ustar (below Eq. 1) has a typo.*

Corrected.

9. *Page 6, line 18: "simplified version": a Reynolds-averaged (RANS) or Filtered (LES) version of the N-S equation is not a "simplified" version.*

We changed the wording to filtered.

10. *Page 6, line 20: "time averaged" should be replaced by "ensemble averaged"*

On advice of the other reviewer we have changed the time average operator into an overbar to reflect the time averaging deployed in RANS models and subsequently used the operator <> to represent the filering in the LES simulations. We have considered the option of using a different operator for ensemble and time average, but since the only real ensemble we have is the measurement data, we have used the same notation for simplicity.

11. *Page 7, line 2: "fully resolved" -> atmospheric LES runs are rarely fully resolved. One has to demonstrate that everywhere in the domain, the subgrid-scale energy is less than 20% of total energy*

The question of fully resolved is complicated, and there are certainly other tests than the 20% as well (see Davidsson 2009). The sentence in the manuscript where *fully resolved* occurs refers to general LES simulations and not the specific ones to this study, hence we would like to keep the sentence as it is. The discussion becomes however important for the question of resolution vs domain size, and we have added a paragraph on that in the discussion.

Davidsson, L, 2009, Large Eddy Simulations: How to evaluate resolution, International Journal of Heat and Fluid Flow, 30-5,1016–1025. doi = "https://doi.org/10.1016/j.ijheatfluidflow.2009.06.006".

12. *Table 1: PALM uses Deardorff's closure. This is a $k_{SGS}$ closure.*

We agree and changed it accordingly.

13. *Page 8, line 16: "limiting maximum length" -> change to "limiting mixing length" or "asymptotic mixing length".*

Corrected.

14. *Page 11, line 13: "et al (2016)" -> missing author.*

Corrected.

15. *Page 11, line 18: What is a "reactive term" in turbulence equations?*

We have reformulated the text avoiding the confusion related to notation (advection/convection and reaction production and dissipation). We hope the new formulation is more clear.

The comments from the reviewer have certainly helped us to improve our manuscript and we hope that the comments have been taken into consideration satisfactorily.

[revised manuscript text omitted]